# Phosphatidylcholines from *Pieris brassicae* eggs activate an immune response in Arabidopsis

**Elia Stahl[1], Théo Brillatz[2,3], Emerson Ferreira Queiroz[2,3], Laurence Marcourt[2,3], André Schmiesing[1], Olivier Hilfiker[1], Isabelle Riezman[4], Howard Riezman[4], Jean-Luc Wolfender[2,3], Philippe Reymond[1]***

[1]Department of Plant Molecular Biology, University of Lausanne, Lausanne, Switzerland; [2]School of Pharmaceutical Sciences, University of Geneva, Geneva, Switzerland; [3]Institute of Pharmaceutical Sciences of Western Switzerland, Geneva, Switzerland; [4]NCCR Chemical Biology, University of Geneva, Geneva, Switzerland

**Abstract** Recognition of conserved microbial molecules activates immune responses in plants, a process termed pattern-triggered immunity (PTI). Similarly, insect eggs trigger defenses that impede egg development or attract predators, but information on the nature of egg-associated elicitors is scarce. We performed an unbiased bioactivity-guided fractionation of eggs of the butterfly *Pieris brassicae*. Nuclear magnetic resonance (NMR) spectroscopy and mass spectrometry of active fractions led to the identification of phosphatidylcholines (PCs). PCs are released from insect eggs, and they induce salicylic acid and $H_2O_2$ accumulation, defense gene expression and cell death in *Arabidopsis*, all of which constitute a hallmark of PTI. Active PCs contain primarily C16 to C18-fatty acyl chains with various levels of desaturation, suggesting a relatively broad ligand specificity of cell-surface receptor(s). The finding of PCs as egg-associated molecular patterns (EAMPs) illustrates the acute ability of plants to detect conserved immunogenic patterns from their enemies, even from seemingly passive structures such as eggs.

**\*For correspondence:** philippe.reymond@unil.ch

**Competing interests:** The authors declare that no competing interests exist.

## Introduction

In nature, plants are frequently confronted with microbial pathogens or herbivores, and thus have evolved an efficient immune response that mainly relies on initial detection of the attacker, followed by production of defense proteins and toxic metabolites (*Jones and Dangl, 2006*; *Schuman and Baldwin, 2016*). What plants perceive is the presence of conserved microbe-associated molecular patterns (MAMPs) and activate pattern-triggered immunity (PTI) (*Ranf, 2017*). In bacteria, flagellin, peptidoglycan or medium-chain 3-hydroxy-fatty acids constitute well-studied sources of MAMPs that are sensed by specific plant cell-surface receptor-like kinases; the same is true for chitin, a structural component of fungal cell walls (*Boutrot and Zipfel, 2017*; *Ranf, 2017*; *Kutschera et al., 2019*).

Similarly, oral secretions (OS) from feeding insect larvae contain herbivore-associated molecular patterns (HAMPs) that trigger defense responses (*Wu and Baldwin, 2010*; *Erb and Reymond, 2019*). Despite the finding that OS from different insect species activate plant defenses, only a few HAMPs have been characterized chemically (*Stahl et al., 2018*). Volicitin, for example, is a fatty acid-amino acid conjugate from several chewing herbivores OS and triggers the release of plant volatile organic compounds that attract parasitic wasps (*Alborn et al., 1997*). Inceptin is a proteolytic fragment of a plant chloroplast ATPase found in *Spodoptera frugiperda* OS and induces the production of volatiles and defense compounds (*Schmelz et al., 2006*). However, contrary to MAMPs, receptors for HAMPs have not yet been described.

Although a seemingly harmless developmental stage of herbivores, insect eggs trigger efficient plant defenses that include necrosis, callus formation, accumulation of ovicidal compounds and release of volatiles to attract egg predators (*Reymond, 2013*; *Hilker and Fatouros, 2015*). The large white butterfly *Pieris brassicae* deposits batches of 20–30 eggs onto *Arabidopsis* leaves, causing a large transcriptional reprogramming that is drastically distinct from the expression profile triggered by larval feeding (*Little et al., 2007*). Also, *P. brassicae* eggs induce localized cell death, accumulation of reactive oxygen species (ROS) and salicylic acid (SA), and expression of PTI-related genes, suggesting that egg-associated molecular patterns (EAMPs) activate a response that is similar to the response induced by microbial pathogens (*Little et al., 2007*; *Bruessow et al., 2010*; *Gouhier-Darimont et al., 2013*). Studies with other Brassicaceae reported localized necrosis, SA accumulation and defense gene expression upon *P. brassicae* oviposition or egg extract (EE) treatment (*Bruessow and Reymond, 2007*; *Fatouros et al., 2008*; *Fatouros et al., 2014*; *Bonnet et al., 2017*; *Griese et al., 2017*). In *Brassica nigra*, variation in the intensity of localized cell death was negatively correlated with egg survival (*Fatouros et al., 2014*; *Griese et al., 2017*). In addition, *Arabidopsis* displays variation in the strength of egg-induced necrosis between accessions, but whether this is sufficient to reduce egg survival needs to be evaluated (*Reymond, 2013*).

So far, only a few EAMPs have been identified and they were found in secretions associated with eggs or in adults. Bruchins are C22-C24 long-chain α,γ-diols esterified at one or both ends with 3-hydroxypropanoic acid. They are present in bodies of cowpea weevil and induce cell division in pea pods, creating a neoplastic tissue that presumably impedes larval entry (*Doss et al., 2000*). Accessory reproductive gland (ARG) secretions covering eggs of *P. brassicae* trigger arrest of the parasitoid wasp *Trichogramma brassicae* on *Brassica oleracea* and *Arabidopsis* by modifying leaf surface chemistry (*Fatouros et al., 2008*; *Blenn et al., 2012*). This indirect defense response is activated by benzyl cyanide and indole, both male-derived anti-aphrodisiacs found in ARG secretions of *P. brassicae* and *P. rapae*, respectively (*Fatouros et al., 2008*; *Fatouros et al., 2009*). Proteins or peptides in oviduct secretions from the pine sawfly *Diprion pini* and the elm leaf beetle *Xanthogaleruca luteola* are responsible for oviposition-induced volatile emission in pine needle and elm leaves, respectively, but their sequence needs to be characterized (*Meiners and Hilker, 2000*; *Hilker et al., 2005*).

Currently, the nature of EAMPs that induce immune responses in *Arabidopsis* is unknown. We previously reported that a crude *P. brassicae* EE (soluble fraction from crushed eggs) induced similar responses as oviposition, including ROS and SA accumulation, cell death and defense gene induction (*Little et al., 2007*; *Bruessow et al., 2010*; *Gouhier-Darimont et al., 2013*). The eggshell was not active implying that gene-induction activity is not associated with covering secretions but is contained in the egg (*Bruessow et al., 2010*). Here, we show that phosphatidylcholines (PCs) from *P. brassicae* eggs trigger SA accumulation and immune responses. We postulate that PCs represent *bona fide* EAMPs and that plants have evolved receptors to perceive an early stage of insect attack.

## Results

### Purification of *P. brassicae* eggs

In order to find marker genes that we could robustly use to identify defense-inducing compounds in eggs, we performed an RNA sequencing experiment using *Arabidopsis* plants on which eggs were naturally oviposited by *P. brassicae* butterflies and compared it with plants treated with EE. After 5 days, hundreds of genes were significantly upregulated by each treatment and their induction was highly similar between treatments (*Figure 1—figure supplement 1*, *Supplementary file 1*). This conserved transcriptomic signature strongly supports our previous observations that oviposition and EE treatment trigger comparable responses in *Arabidopsis*. Amongst the most highly upregulated genes, we selected *PATHOGENESIS-RELATED PROTEIN1* (*PR1*, At2g14610), *SENESCENCE-ASSOCIATED GENE 13* (*SAG13*, At2g29350), and *KUNITZ INHIBITOR PROTEIN 1* (*TI*, At1g73260), which were equally induced by oviposition and EE treatment (*Figure 1—figure supplement 1*, *Supplementary file 1*) and which were initially found to be strongly responsive to *P. brassicae* oviposition and to treatment by EE from different insect species (*Bruessow et al., 2010*). Furthermore, *PR1* is a known marker gene of the SA pathway (*van Loon et al., 2006*), whereas *SAG13* and *TI* have

been shown to be involved in the regulation of cell death and defense (*Brodersen et al., 2002*; *Li et al., 2008*; *Dhar et al., 2020*).

Preliminary tests indicated that egg-derived defense eliciting compounds are of lipidic nature (*Bruessow et al., 2010*; *Gouhier-Darimont et al., 2013*). To confirm this observation, we used Clea-nascite solid-phase aqueous reagent to selectively adsorb lipids from EE. Application of the lipid-containing phase to PR1::GUS, SAG13::GUS, and TI::GUS *Arabidopsis* reporter lines triggered strong and localized GUS staining, similar to EE treatment. In contrast, the supernatant containing proteins and other non-lipidic molecules was not active, indicating that defense gene-inducing mole-cules were restricted to the lipid phase (*Figure 1A*). Then, we collected approximately 100'000 *P. brassicae* eggs and extracted 1.1 g of total lipids with $CHCl_3$/EtOH (1:1, v/v). The lipid fraction (LF) was separated onto a solid-phase extraction (SPE) C18-cartridge by elution with increasing concen-tration of MeOH followed by a final wash with ethyl acetate. Each fraction was tested for its ability to induce *PR1* expression by qPCR and compared to the defense-inducing capability of LF. Most of the inducing activity was found in the fraction that eluted with 100% MeOH (Fr. 4, *Figure 1B*). Fr. four was further separated by reverse-phase semi-preparative HPLC with evaporative light scattering detector (ELSD) and 17 subfractions (Fr. 4.1 to Fr. 4.17) that corresponded to peaks eluting during the isocratic phase of the gradient (see Materials and methods) were collected and tested for activ-ity. Subfractions Fr. 4.10 to 4.17 induced *PR1* expression, Fr. 4.14, Fr. 4.15, and Fr. 4.16 being the most active ones (*Figure 1C*).

## Identification of active PCs

To obtain information about the chemical composition of active fractions, we used one-dimensional (1D) and two-dimensional (2D) $^1$H and $^{31}$P nuclear magnetic resonance (NMR) spectroscopy. The $^1$H NMR spectrum of Fr. four displayed typical profile of phosphatidylcholine (PC) derivatives with sig-nals from the glycerol part at $\delta_H$ 5.24, 4.44, 4.18, 4.00, and from the choline part at $\delta_H$ 4.28, 3.65, 3.23. The $^{31}$P NMR spectrum confirmed the presence of phosphorylated compounds and indicated that it contained different classes of phospholipids (*Figure 1—figure supplement 2*). To identify them, a 2D $^1$H−$^{31}$P heteronuclear total correlation spectroscopy (TOCSY) experiment was per-formed (*Balsgart et al., 2016*). Then, $^{31}$P signals (each corresponding to a class of phospholipid) from each subfraction was integrated and the concentration was calculated using a solution of 48.5 mM of triphenyl phosphate as external standard. The active subfractions Fr. 4.10–4.17 contained mainly phosphatidylcholines (PCs), with low amounts of sphingomyelin (SM) and phosphatidyletha-nolamines (PEs) (*Figure 2A*). To confirm these findings, a shotgun lipidomics analysis of *P. brassicae* EE by direct infusion mass spectrometry (DIMS) (*Surma et al., 2015*) indicated that PCs are the most abundant lipid species (46%), followed by PEs (23%), triacylglycerides (14%), diacylglycerides (7%), and other lipids at lower concentrations (*Figure 2B*). PC species contained mainly C16 to C18 fatty acyl chains with different combinations and levels of desaturation (*Figure 2C*). For clarity, we employ the following PC nomenclature: when known, the length and level of desaturation of each of the two fatty acyl chain is indicated (e.g., PC(18:1/18:1)); otherwise, only the total number of carbons and double bonds is mentioned (e.g., PC36:2); in addition, sn-1 or sn-2 position of different acyl chains is not indicated.

Total PC quantification in *P. brassicae* EE by 1D $^{31}$P NMR or by MS shotgun lipidomics yielded a similar concentration of ca. 5 µg/µL (*Figure 2D*), which corresponds to 0.5 µg of PCs per egg. In addition, a similar lipid content and concentration were found in eggs of the generalist moth *Spo-doptera littoralis* (*Figure 2—figure supplement 1*), which were previously shown to also induce defense responses in *Arabidopsis* (*Bruessow et al., 2010*).

Results from the purification of *P. brassicae* EE clearly suggested that phospholipids, and presum-ably PCs, are active compounds in insect eggs. To test this hypothesis, we treated EE with phospho-lipase D, which preferentially cleaves the choline headgroup of PCs, and phospholipase A2, which cleaves the fatty acid in position sn-2 of phospholipids. A combined treatment with both phospholi-pases abolished *PR1*-inducing activity of EE, confirming that a phospholipid is crucial for triggering defense gene expression (*Figure 2E*). In addition, treatment of EE with single phospholipases showed that PLA2 reduced *PR1*-inducing activity of EE to ca. 20% whereas PLD was less effective, probably because of the release of phosphatidic acid (PA) from PCs (see below) (*Figure 2—figure supplement 2*). Loss of the $^{31}$P NMR signal for PCs and PEs in PLD/PLA$_2$-treated sample indicated that phospholipases efficiently degraded the main lipid classes of EE (*Figure 2F*). To further evaluate

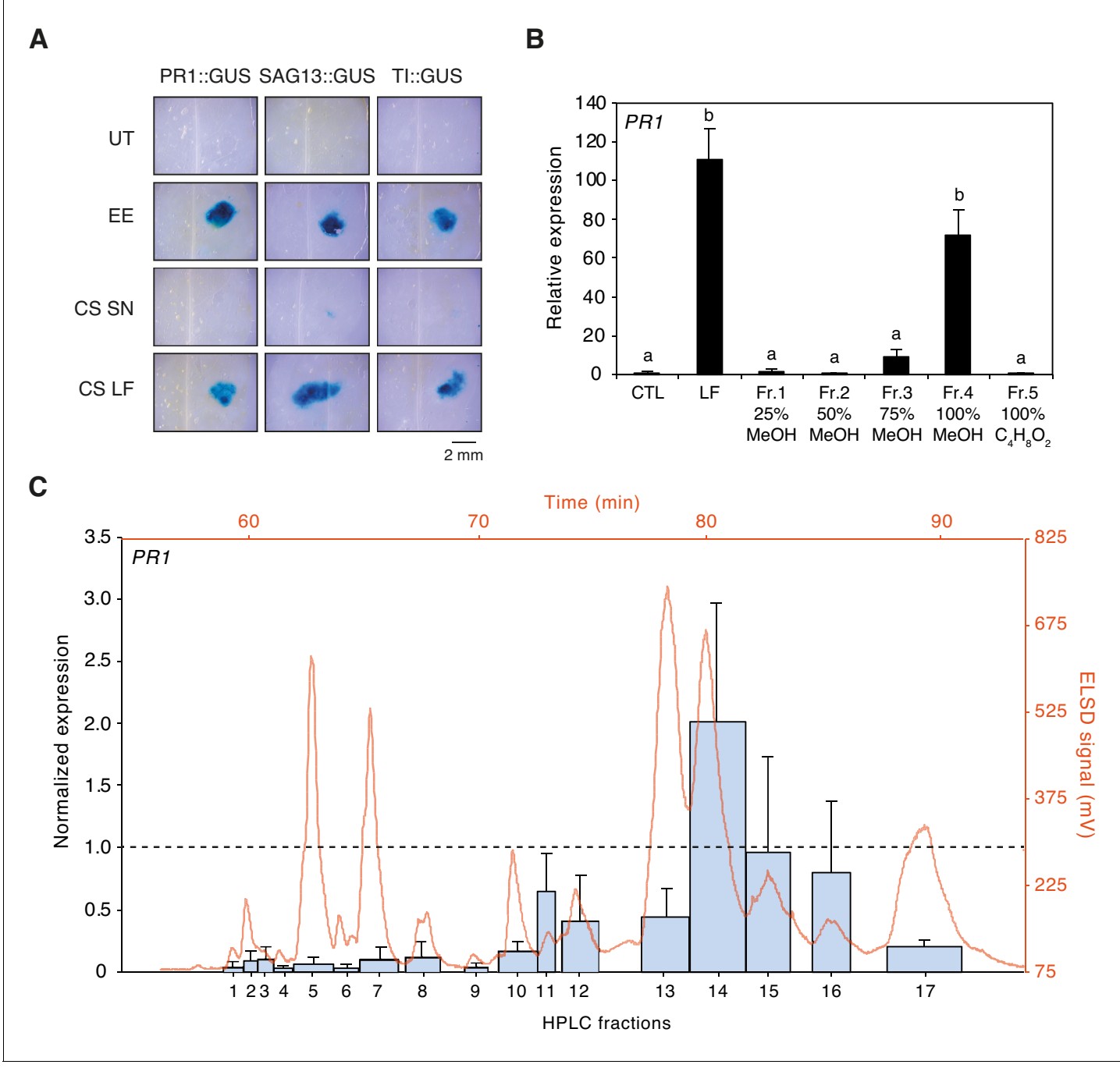

**Figure 1.** Purification of plant-defense eliciting *P. brassicae* egg lipids. (**A**) Expression of defense genes *PR1*, *SAG13*, and *TI* in response to purified *P. brassicae* egg lipids. Purification of egg lipids was conducted using Cleanascite. GUS reporter lines were treated with the lipid-free supernatant (CS SN) or the lipid fraction (CS LF). Untreated and egg extract (EE)-treated plants served as controls. The experiment was repeated three times with similar results and representative pictures from one experiment are shown. (**B**) Relative *PR1* expression upon treatment with purified *P. brassicae* egg lipids (LF) and with fractions from LF separated by solid-phase extraction (SPE). LF and SPE fractions were applied at 5 µg/µL solved in 1% DMSO and plants treated with 1% DMSO served as controls. Transcript levels represent means ± SE of three independent experiments. Different letters indicate significant differences between treatments (ANOVA followed by Tukey's honest significant difference test, p<0.05). (**C**) *PR1* expression upon treatment with fractions obtained from semi-preparative HPLC-fractionation of SPE Fr. four detected in ELSD (mV). Subfractions Fr. 4.1 to Fr. 4.17 were applied at 5 µg/µL solubilized in 1% DMSO. *PR1* expression was normalized to the expression value obtained upon treatment with SPE Fr. four (indicated by the dashed line). Transcript levels represent means ± SE of two to five independent experiments. HPLC chromatogram used for fraction collection is indicated in orange.

The online version of this article includes the following source data and figure supplement(s) for figure 1:

*Figure 1 continued on next page*

*Figure 1 continued*

**Source data 1.** Source data for *Figure 1B and C*.
**Figure supplement 1.** RNAseq analysis of *Arabidopsis* leaves after oviposition or egg extract (EE) treatment.
**Figure supplement 1—source data 1.** Source data for *Figure 1—figure supplement 1*.
**Figure supplement 2.** Two-dimensional $^1H-^{31}P$ TOCSY NMR experiment of SPE Fr. four in $CD_3OD$.

the specific contribution of PCs to the *PR1*-inducing activity of EE, we reconstituted a synthetic egg lipid mix (SELM). We included the major phospholipids identified in EE and active fractions (PC, PE, TAG, DAG, LPC, LPE; *Figure 2A,B*) and added them at their respective concentration. Given the potential role of PA, we also added PA at the concentration found in EE. SELM application triggered *PR1* expression to a similar extent as EE (*Figure 2G*). However, when PCs were omitted from the mix (SELM-PC), no *PR1* expression could be detected, strongly suggesting that PCs are responsible for the observed induction. Indeed, treatment with a PC-Mix triggered *PR1* induction, albeit to a lower extent, which could indicate that some additional EE components are needed for a full PC response or that PCs are more potent when diluted in the EE solution (*Figure 2G*).

## PCs are active molecules in eggs

To obtain further information on the chemical nature of PCs contained in active subfractions, a MS/MS analysis was performed in positive and negative modes and precursor ion scans were analyzed. To identify fatty acyl chains, LiAc was added and loss of specific fatty acids was detected by neutral ion loss analysis. Results indicated the presence of PC(16:0/16:1) (*m/z* 732.39) in Fr. 4.14, PC(18:1/18:3) (*m/z* 782.37) in Fr. 4.13 and Fr. 4.14, PC(16:1/18:1) (*m/z* 758.43) in Fr. 4.14, and PC36:3 (*m/z* 756.40) in Fr. 4.13 (*Figure 2—figure supplement 3*). The fatty acyl chains of PCs in other active sub-fractions could not be determined precisely due to a lack of clear fragmentation pattern. Then, to test whether pure PCs can activate defense gene expression, we applied commercially available phospholipids onto PR1::GUS, SAG13::GUS, and TI::GUS *Arabidopsis* lines. All PCs with C16 or C18 fatty acyl chains, including PC(16:1/16:1) found in very small amount in Fr. 4.13, and the abundant PC(18:3/18:3) identified by MS shotgun lipidomics, robustly activated gene expression in the three reporter lines, irrespective of the level of fatty acid desaturation. A mixture of purified PCs from chicken egg yolk was also active. In contrast, a PC with C3 fatty acyl chains, lysoPCs, PEs, lysoPEs, and sphingomyelin (SM) were inactive (*Figure 3A*). Then, to test which part of the PC molecule is important for activity, we applied different PC constituents to GUS reporter lines. The entire PC(18:1/18:1) molecule was active, as well as PA (which lacks the choline head group), whereas choline, phosphocholine, lysophosphocholine LPC(18:1/18:1) or 18:1 free fatty acid did not cause GUS staining (*Figure 3B*). We further evaluated the *PR1*-inducing activity of phospholipids found in EE. At the same concentration (1 µg/µL), PC(18:1/18:1) robustly activated *PR1* gene expression whereas LPC(18:1), PE(18:1/18:1), LPE(18:1), DAG(16:0/18:1), and TAG(18:1/18:1/18:1) were inactive. As observed with GUS lines, PA(18:1/18:1) activated *PR1* gene expression as well but to a lower extent than PC(18:1/18:1) (*Figure 3—figure supplement 1*). Finally, treatment with 10 µg of PC(16:1/16:1), which corresponds to the amount of total PC found in 2 µL of EE used for experiments, triggered *PR1* expression to similar levels as EE treatment. However, lower amounts down to 0.02 µg (13.6 µM) still significantly upregulated *PR1* expression, indicating that PCs are active at low micromolar range (*Figure 3C*). Overall, these data indicate that PCs containing long- chain fatty acids are active molecules in insect eggs, but that the type of fatty acid desaturation and exact length are not crucial for activity.

## PCs can diffuse out of the eggs

Having shown that PCs are abundant components of *P. brassicae* and *S. littoralis* eggs, that they are found in *PR1*-inducing fractions and that they can activate defense gene expression when applied exogenously, we wondered if phospholipids are released by eggs onto the leaf after oviposition. To tackle this technically challenging question, we took advantage of the ability of butterflies to lay eggs on filter paper. Eggs from *P. brassicae* and *S. littoralis* deposited on filter paper were gently removed after one or three days, respectively, and the presence of phospholipids on the filter paper was measured by MS/MS. Strikingly, PCs were identified on filter paper and their profile was highly

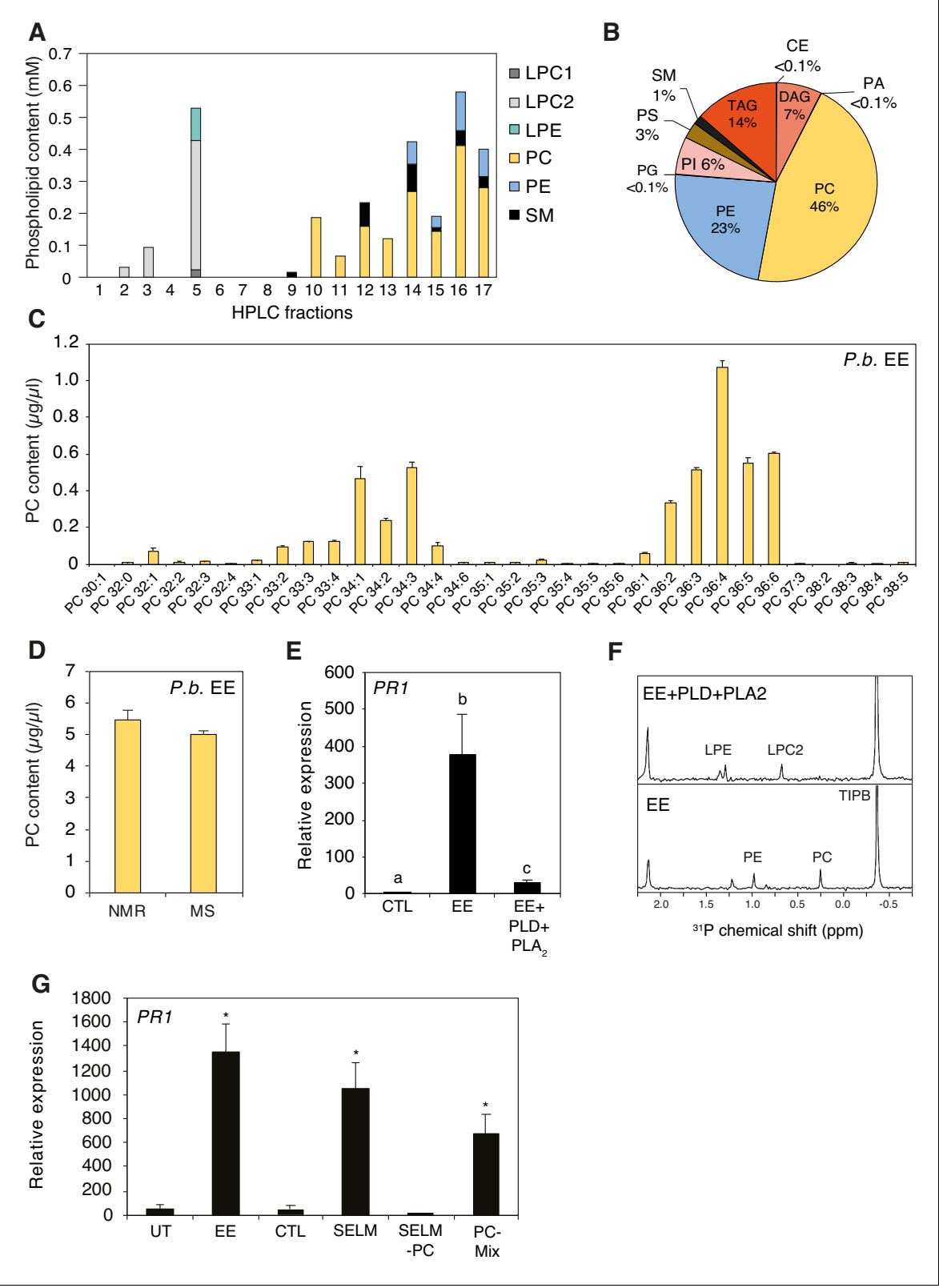

**Figure 2.** Identification of phospholipids as plant-defense eliciting compounds in *P. brassicae* eggs. (**A**) Phospholipid composition of HPLC fractions was obtained by 1D $^{31}$P NMR. (**B**) Lipid composition of *P. brassicae* EE measured by MS shotgun lipidomic analysis. Values are given as percentage of the total lipid mass and display means of three different EE preparations. (**C**) Absolute levels of phosphatidylcholines (PCs) in *P. brassicae* EE measured by MS shotgun lipidomic analysis. Different PC species are reported according to their molecular composition with the total number of carbon atoms

*Figure 2 continued on next page*

*Figure 2 continued*

and the sum of double bonds in fatty acyl chains. PC levels represent the means ± SE of three different EE preparations. (**D**) Absolute quantification of total PC content in *P. brassicae* EE measured by 1D $^{31}$P NMR and MS shotgun lipidomic analysis. Total PC levels represent means ± SE of three different EE preparations. (**E**) Relative *PR1* expression upon treatment with EE and EE digested with phospholipase D (PLD) and phospholipase A$_2$ (PLA$_2$). Untreated plants served as controls (CTL). Transcript levels represent means ± SE of three independent experiments. Different letters indicate significant differences between treatments (ANOVA followed by Tukey's honest significant test, p<0.05). (**F**) 1D $^{31}$P NMR spectra of EE + TIPB (bottom) and EE + PLD + PLA$_2$ + TIBP (top). TIBP is used as an internal standard. (**G**) Relative *PR1* expression upon treatment with EE, a synthetic egg lipid mix (SELM), a synthetic egg lipid mix without PCs (SELM-PC), and a PC-Mix. The SELM was composed of 5 µg/µL PC-Mix, 2.5 µg/µL PE-Mix, 1.75 µg/µL TAG(18:1/18:1/18:1), 1 µg/µLDAG(16:0/18:1), 1 µg/µL LPC-Mix, 0.2 µg/µL LPE-Mix and 0.1 µg/µL PA-Mix. The compounds were solubilized in 1% DMSO, 0.5% Glycerol and 0.1% Tween. Plants treated with 1% DMSO, 0.5% Glycerol and 0.1% Tween (CTL) and untreated plants (UT) served as controls. Transcript levels represent means ± SE of three independent experiments. Asterisks denote statistically significant differences between control and treated plants (Welch's *t*-test, *p<0.05). CE, cholesterol ester; DAG, diacylglycerol; LPC1 and LPC2, lysophosphatidylcholine; LPE, lysophosphatidylethanolamine; PA, phosphatidic acid; PC, phosphatidylcholine; PE, phosphatidylethanolamine; PG, phosphatidylglycerol; PI, phosphatidylinositol; PS, phosphatidylserine; SM, sphingomyelin; TAG, triacylglycerol.

The online version of this article includes the following source data and figure supplement(s) for figure 2:

**Source data 1.** Source data for *Figure 2A–G*.
**Figure supplement 1.** Comparative lipid composition of *P. brassicae* and *S. littoralis* EE.
**Figure supplement 1—source data 1.** Source data for *Figure 2—figure supplement 1A–C*.
**Figure supplement 2.** Relative *PR1* expression upon treatment with EE and EE-treated with phospholipase A$_2$ (PLA$_2$) or phospholipase D (PLD).
**Figure supplement 2—source data 1.** Source data for *Figure 2—figure supplement 2*.
**Figure supplement 3.** Precursor ion scans for choline-containing lipids in active fractions.

similar to the PC profile found in EE from both insects (*Figure 4*). For example, PC(36:2) to PC(36:6) and PC(34:1) to PC(34:3) were the most abundant PCs from *P. brassicae* EE and filter paper samples. In contrast, empty control filter paper contained low levels of PCs (*Figure 4—figure supplement 1*). The total amount of PCs released by *P. brassicae* eggs during one day was 0.23 ± 0.05 µg / batch, which falls in the range of active concentrations that induce *PR1* expression (*Figure 3C*). *S. littoralis* eggs released four-fold more PCs (*Figure 4*), which can be explained by a higher number of eggs deposited and a longer exposure to the filter paper. We thus conclude that PCs can diffuse out of the eggs and are released in sufficient amounts to trigger a response in the plant.

## PCs and EE induce similar defense responses

Oviposition or EE application trigger similar SA accumulation, ROS production, cell death, and defense gene expression in *Arabidopsis* (*Little et al., 2007*; *Bruessow et al., 2010*; *Gouhier-Darimont et al., 2013*; *Gouhier-Darimont et al., 2019*; *Figure 1—figure supplement 1*). In addition, we recently showed that the receptor-like kinase LecRK-I.8 is an early component of egg-induced signaling responses (*Gouhier-Darimont et al., 2019*). To test whether PCs can mimic these responses in a LecRK-I.8-dependent manner, *Arabidopsis* Col-0 and a T-DNA knock-out *lecrk-I.8* mutant were exposed to oviposition by *P. brassicae* butterflies, or treated with EE and different synthetic PCs. Eggs, EE or PCs induced similar SA accumulation, and this response was significantly reduced in *lecrk-I.8* (*Figure 5A*). Then, although they were not as potent as oviposition or EE in triggering defense gene expression, a PC-Mix, PC(16:1/16:1), and PC(18:3/18:3) induced *PR1*, *SAG13* and *TI* expression in Col-0, but significantly less in *lecrk-I.8* (*Figure 5B*). In addition, to test if these immune responses could be triggered by a release of leaf-derived PCs that would act as damage-associated molecular patterns (DAMPs), we gently wounded the abaxial leaf with forceps on a surface equivalent to the EE- or PC-treated area. Unlike oviposition, EE or PC treatment, wounding did not trigger SA accumulation and defense gene expression, strongly suggesting that the observed effects are due to egg-derived PCs (*Figure 5A,B*). Finally, like EE treatment, PCs triggered local accumulation of H$_2$O$_2$ and cell death, and the response was significantly lower in *lecrk-I.8* (*Figure 5C,D,E*). Thus, these data show that PCs closely mimic the effect of EE in inducing PTI-like responses, likely through LecRK-I.8 activity, supporting their role as EAMPs from insect eggs.

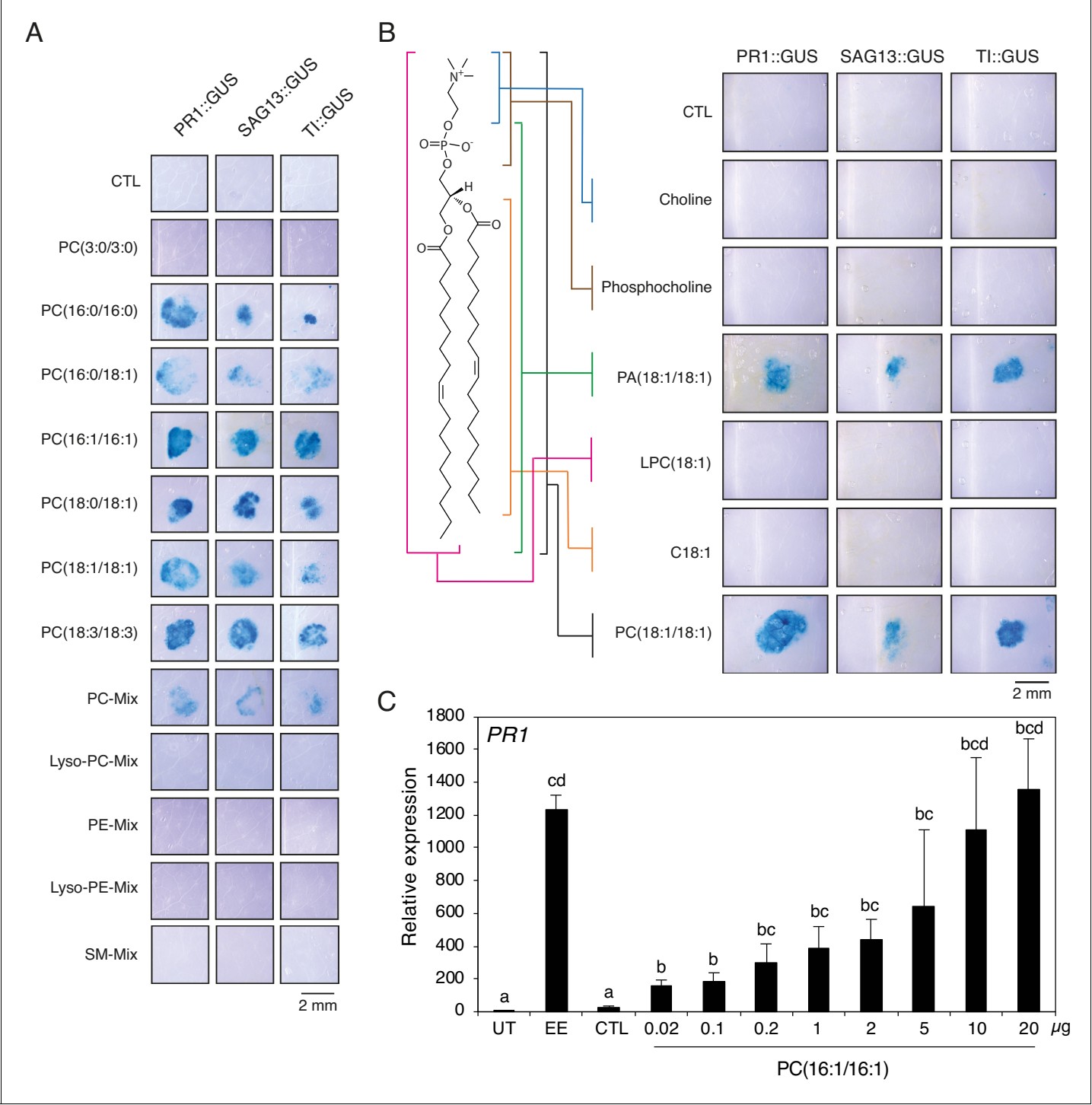

**Figure 3.** Extracellular phosphatidylcholines activate defense gene expression. (A) Expression of *PR1*, *SAG13*, and *TI* upon treatment with different PCs and purified phospholipid preparations. Phospholipids were applied to GUS reporter lines at 1 µg/µL, solubilized in 1% DMSO, 0.5% Glycerol and 0.1% Tween. Plants treated with 1% DMSO, 0.5% Glycerol and 0.1% Tween (CTL) served as controls. Each phospholipid was tested at least twice for its capability to activate defense gene expression and representative pictures from one experiment are shown. (B) Expression of *PR1*, *SAG13*, and *TI* in response to treatment with choline, phosphocholine, phosphatidic acid (PA) (18:1/18:1), lysophosphatidylcholine LPC(18:1), oleic acid (C18:1), and PC (C18:1/C18:1). All compounds were applied to GUS reporter lines at 1 µg/µL, solubilized in 1% DMSO, 0.5% Glycerol and 0.1% Tween. Plants treated with 1% DMSO, 0.5% Glycerol and 0.1% Tween (CTL) served as controls. The experiment was repeated three times with similar results and representative pictures from one experiment are shown. (C) Relative *PR1* expression upon treatment with EE and PC(C16:1/C16:1) at different concentrations. PC(C16:1/C16:1) was solubilized in 1% DMSO, 0.5% Glycerol and 0.1% Tween. Untreated plants (UT) and plants treated with 1% DMSO,

*Figure 3 continued on next page*

*Figure 3 continued*

0.5% Glycerol and 0.1% Tween (CTL) served as controls. Total amounts of PC(C16:1/C16:1) applied per treatment are given below each bar. Transcript levels represent means ± SE of four independent experiments. Different letters indicate significant differences between treatments (ANOVA followed by Tukey's honest significant test, p<0.05). SM-mix, sphingomyelin mix.

The online version of this article includes the following source data and figure supplement(s) for figure 3:

**Source data 1.** Source data for *Figure 3C*.

**Figure supplement 1.** PC(18:1/18:1) and PA(18:1/18:1) activate defense gene expression in Arabidopsis.

**Figure supplement 1—source data 1.** Source data for *Figure 3—figure supplement 1*.

## Discussion

We show here that purified PCs from *P. brassicae* eggs and synthetic PC species trigger SA and ROS accumulation, local cell death, and defense gene induction in *Arabidopsis*. Since these responses are also observed after oviposition, our findings thus support the conclusion that PCs

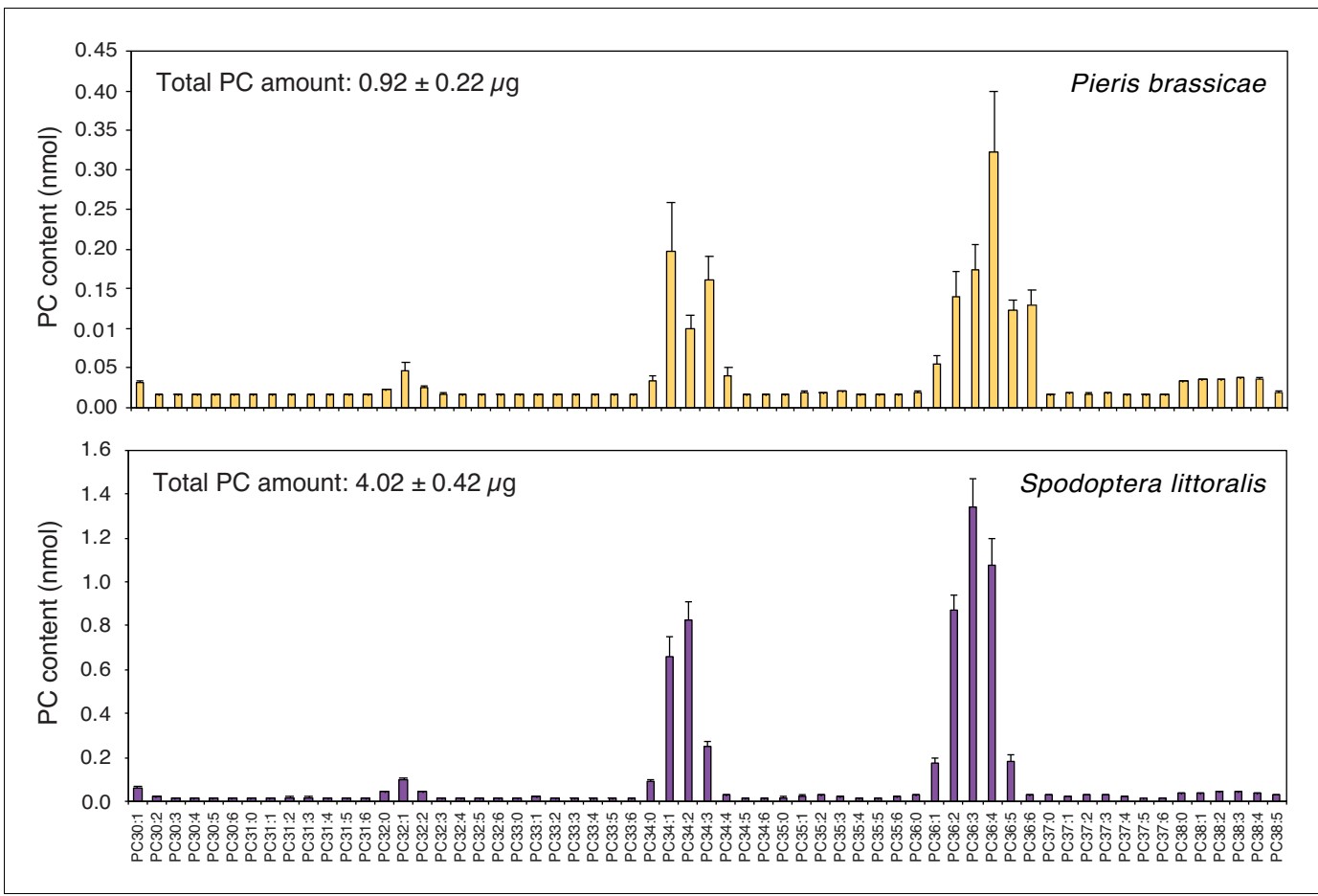

**Figure 4.** Phosphatidylcholines are released from intact insect eggs. MS-based quantification of PCs extracted from filter papers on which *P. brassicae* (upper panel) and *Spodoptera littoralis* (lower panel) eggs were deposited by natural oviposition. 80 *P. brassicae* eggs or 150–200 *s. littoralis* eggs were oviposited on filter papers and PCs from the filter papers were extracted one day and three days later, respectively. PC species are reported according to their molecular composition with the total number of carbon atoms and the sum of double bonds in the fatty acyl chains and the levels represent the mean ± SE of three independent samples. Total amount of PCs is describing the sum of all PC species per filter paper extract and represents the mean ± SE of three independent samples.

The online version of this article includes the following source data and figure supplement(s) for figure 4:

**Source data 1.** Source data for *Figure 4*.

**Figure supplement 1.** A full MS scan of samples extracted from filter paper (**A**) control (no eggs) and *P. brassiceae* eggs oviposited on filter paper and removed after one day.

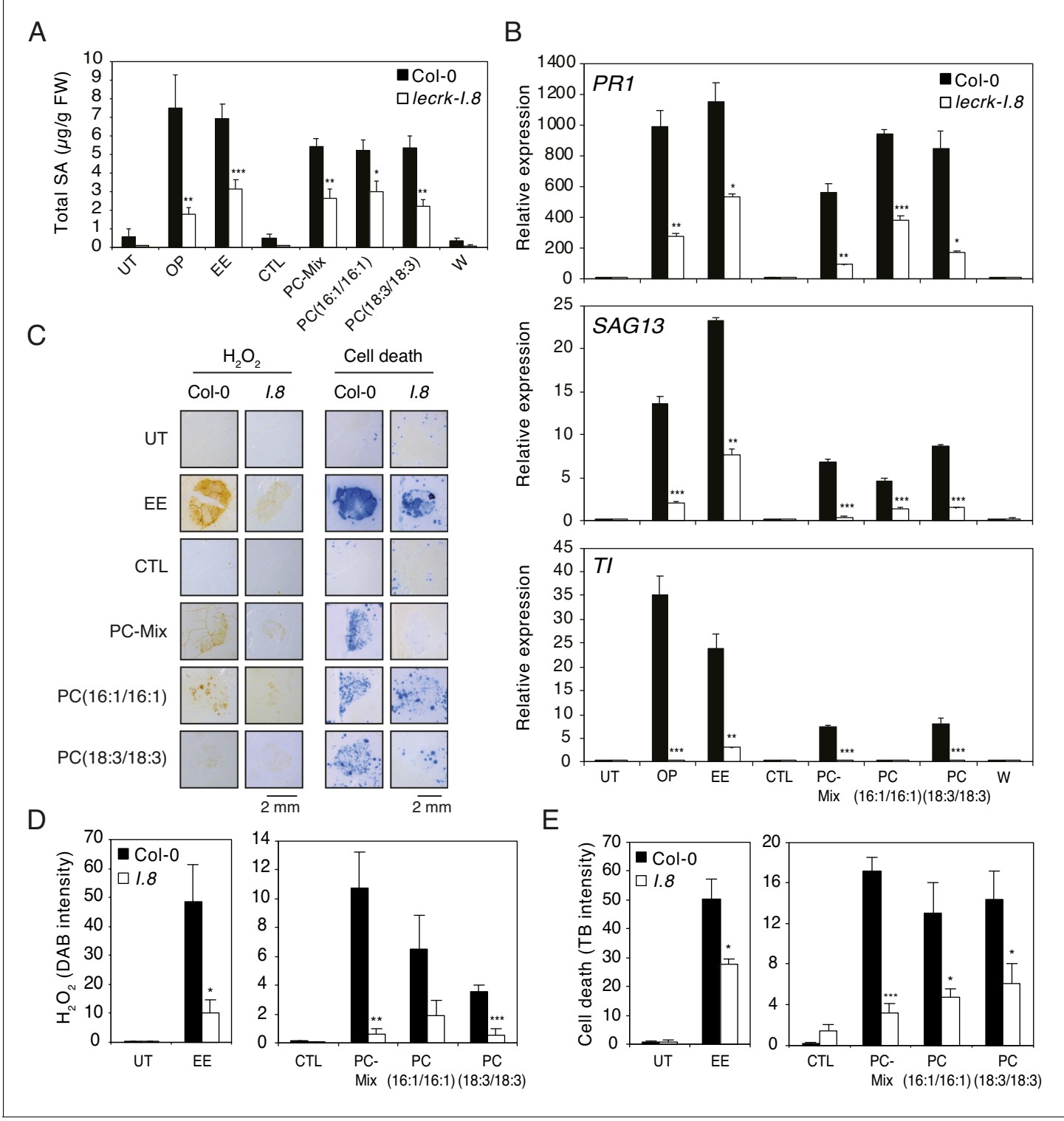

**Figure 5.** Elicitation of plant-defense responses by extracellular phosphatidylcholines depends on a functional Lectin Receptor Kinase LecRK-I.8. (**A**) Endogenous levels of salicylic acid (SA) in Col-0 and *lecrk-I.8* in response to *P. brassicae* oviposition (OP), EE, PC-Mix, PC(C16:1/C16:1), PC(C18:3/C18:3), and wounding (W). Total SA levels represent means ± SE of four independent experiments. Asterisks denote statistically significant differences between the same treatment of Col-0 and *lecrk-I.8* (Welch's *t*-test, ***p<0.001). (**B**) Relative expression of *PR1*, *SAG13* and *TI* in Col-0 and *lecrk-I.8* upon treatment with oviposition (OP), EE, PC-Mix, PC(C16:1/C16:1), PC(C18:3/C18:3), and wounding (W). Transcript levels represent means ± SE of one representative experiment with technical triplicates. The experiment was repeated three times with similar results. Asterisks denote statistically significant differences between the same treatment of Col-0 and *lecrk-I.8* (Welch's *t*-test, *p<0.05, **p<0.01, ***p<0.001). (**C**) Histochemical staining of

*Figure 5 continued on next page*

*Figure 5 continued*

leaves of Col-0 and *lecrk-I.8* with 3,3-diaminobenzidine (DAB) to detect $H_2O_2$ accumulation and trypan blue to detect cell death in response to treatment with EE, PC-Mix, PC(C16:1/C16:1), and PC(C18:3/C18:3). The experiment was repeated twice with similar results and representative pictures from one experiment are shown. (D, E) Quantification of $H_2O_2$ and cell death accumulation as in (C). Staining intensity was measured on images with ImageJ software. Means ± SE of five to six leaves are shown. Asterisks denote statistically significant differences between the same treatment of Col-0 and *lecrk-I.8* (Mann-Whitney *U* test (D) and Welch's *t*-test (E), *, $p<0.05$; **, $p<0.01$; ***, $p<0.001$). For all experiments, PCs were applied at 1 µg/µL (A, B) or 5 µg/µL (C–E), solubilized in 1% DMSO, 0.5% Glycerol and 0.1% Tween. Untreated plants (UT) and plants treated with 1% DMSO, 0.5% Glycerol and 0.1% Tween (CTL) served as controls.

The online version of this article includes the following source data for figure 5:

**Source data 1.** Source data for *Figure 5A–E*.

contribute significantly to the biological responses to eggs. Identification of active PCs in eggs of chewing herbivores contributes to a relatively small list of characterized EAMPs and, to the best of our knowledge, represents the only example of EAMPs that are clearly shown to originate from the egg itself. PCs are commonly found in storage lipids of insect eggs and are major components of biological membranes (*Bridges, 1972*). Thus, like MAMPs and HAMPs, they constitute a classical example of conserved molecules present in whole classes of attackers with an essential function for these attackers (*Boller and Felix, 2009*). The observation that intact eggs or application of crude EE without wounding induce immune responses implies that PCs originate from the eggs, are released at the leaf surface and reach the extracellular space. In support of this hypothesis, we show that eggs deposited on filter paper release significant amounts of PCs. The amphiphilic nature of PCs is likely to allow diffusion through the lipidic cuticle layer and then through the hydrophilic cell wall. In addition, wounding of a leaf is not sufficient to trigger the immune responses observed after oviposition or PC application, excluding potential leaf-derived PCs as the source of inducing activity. The intact PCs and, to a lesser extent PA, induced *PR1* expression but not lysoPCs, choline, phosphocholine, and 18:1 free fatty acid. PA is a known endogenous signal in plant defense and can be generated from PC by PLD activity (*Lim et al., 2017*). We found that PA induced *PR1* expression at 1 µg/µL, which is about 50x more than the PA concentration measured in EE. However, when PA was included at the concentration found in eggs in a synthetic lipid mix lacking PC, the mix did not induce *PR1*. We therefore conclude that PCs are the active molecules in eggs and that they may be further processed to PA in the plant. Whether this would be due to extracellular or cytoplasmic PLD activity seems unlikely since we showed that PE, which can also yield PA upon PLD hydrolysis, is not active in inducing defense gene expression. Alternatively, since the immune response induced by eggs is quite similar to the response induced by microbial pathogens, the observation that PA induces *PR1*, *SAG13* and *TI* genes may be linked to its known signaling role during pathogenesis and not related to PC activity in response to oviposition. Further research will need to address this hypothesis.

PC levels found in a typical *P. brassicae* egg batch of 20 eggs correspond to ca. 10 µg (0.5 µg/egg) and application as low as 0.02 µg of PC(16:1/16:1) induced *PR1* expression. Although the exact amount of egg-derived PCs that reach the plant extracellular space is unknown, we showed that one *P. brassicae* egg batch released 0.23 µg in one day, indicating that, if this is due to a passive diffusion process, up to 1 µg can reach the plant surface after 5 days when eggs hatch. These values thus indicate that the PC amounts present at the plant surface are within the range of quantities that were shown to activate defenses. In addition, the intensity of *Arabidopsis* responses to PC application was not always as strong as with EE treatment. As EE contains several active PCs, synergistic effects of different PCs cannot be excluded. Alternatively, EE may contain additional factors (lipid carriers, antioxidants, adjuvant) that maintain PC stability and facilitate penetration through the plant cuticle and cell wall.

When released in the extracellular space, PCs most likely interact with a plasma-membrane anchored receptor, like all presently known MAMPs (*Boutrot and Zipfel, 2017*). Interestingly, we show here that oviposition, EE and PC treatment induce PTI-like responses and that these responses are drastically reduced in *lecrk-I.8*. It is thus tempting to speculate that LecRK-I.8, a member of clade I L-type LecRKs, is involved in the direct perception of PCs originating from *P. brassicae* eggs. However, the observation that PTI response is not fully abolished in *lecrk-I.8* suggests some level of redundancy. This is supported by the fact that the 11 highly homologous clade I LecRKs are found in

two clusters in the *Arabidopsis* genome, rendering generation of higher order mutants technically challenging (*Bellande et al., 2017*). Alternatively, LecRK-I.8 may participate in a downstream amplification step of the egg-derived signaling pathway. Clearly, future work will be needed to understand how PCs are released from insect eggs and reach the cell surface. Moreover, binding assays with recombinant proteins will be necessary to test the role of LecRK-I.8 and related homologues.

*Arabidopsis* responded to different PCs, but not to PEs or lysoPCs/PEs, indicating some level of specificity. Plant leaf lipids are the major source of fatty acids for Lepidoptera and these fatty acids are then incorporated into neutral lipids and phospholipids in adults and eggs (*Turunen, 1974*; *Turunen, 1990*; *Blomquist et al., 1991*). The dominant fatty acids found in *P. brassicae* eggs from females fed on *Brasssica oleracea* are C16 and C18 with 0 to 3 double bonds (*Turunen, 1974*), and this is consistent with our shotgun MS analysis of PCs in EE. With regard to the position of double bonds, the unsaturated fatty acyl moieties of PCs identified in *P. brassicae* eggs are thus likely 18:1 (n-9), 18:2(n-6), 18:3(n-3), and 16:1(n-7), which are fatty acids found in plants (*McConn and Browse, 1996*). It has to be noted at this stage that PCs are challenging to fully resolve by HPLC due to their similar physicochemical properties and full structural characterization is still challenging (*Hancock et al., 2017*). Thus, the exact chemical structure of all egg-derived PCs will await further development of separation and ion activation techniques. However, we found that the relative length (C16 or C18) and level of desaturation do not matter for PC activity and that the specificity of the response is due to the presence of two fatty acyl chains. The ability to recognize diverse egg PCs may be advantageous for plants as it renders the evolution of avoidance of host detection more difficult for insect pests.

In response to oviposition by the planthopper *Sogatella furcifera*, some rice varieties produce the ovicidal compound benzyl benzoate as a direct defense (*Seino et al., 1996*). Interestingly, purification of female *S. furcifera* extracts led to the identification of two active fractions that triggered benzyl benzoate accumulation. The first one contained PC(18:2/18:2) and the second was a mix of PE (16:0/16–0), PE(16:0/18:1), and PE(18:1/18:1) (*Yang et al., 2014*). Since females insert their ovipositor into stem or leaf sheath of the plant to deposit eggs, it is currently unknown whether PCs or PEs derived from the ovipositor, egg-coating secretions, or eggs. Further work should clarify the origin and specificity of active phospholipids in *S. furcifera* and identify downstream rice signaling components. Although we found that PEs did not activate gene expression in *Arabidopsis*, these results nevertheless suggest that PCs activate defense responses in both *Arabidopsis* and rice. Collectively, identification of active PCs in *P. brassicae* and *S. littoralis* eggs and the finding of related compounds in *S. furcifera* females suggests that widely divergent insect species contain similar signals that alert plants about the presence of eggs on their leaves. Furthermore, EEs from *P. brassicae*, *S. littoralis*, *Trichoplusia ni*, and *Drosophila melanogaster* activate *PR1* expression in *Arabidopsis* (*Bruessow et al., 2010*; *Wang et al., 2017*), indicating that EAMPs from different eggs activate a common signaling pathway, potentially through the perception of PCs.

It is intriguing that plants perceive egg PCs as a potential sign of future insect attack, since PCs are also constituents of plant and bacterial membranes. Although one could argue that the detection of egg-derived PCs is likely to occur in the extracellular space by a cell-surface receptor, hence providing a non-self signal to the plant, leaf-derived PCs could also be passively released upon cell damage and perceived as a sign of danger. Such 'self' patterns are referred to as DAMPs and are known to trigger PTI-like responses. Known DAMPs in plants include ATP, NAD(P), and cell wall fragments (*Gust et al., 2017*). A perception system for extracellular PC may have been an ancestral way to detect cell damage, in conjunction with the detection of other DAMPs, which may have been co-opted to respond to oviposition. However, we did not observe SA accumulation and defense gene expression when we wounded leaves on an area equivalent to the site of EE or PC treatment. Thus, whether PCs are found in the extracellular space upon wounding and activate immune responses remains unknown and would need to be further investigated. In addition, unlike EE and PCs, wounding is not known to trigger the SA pathway in *Arabidopsis* (*Pieterse et al., 2012*).

Because of their ubiquitous presence in biological membranes and storage lipids of eukaryotes, PCs from different sources might be detected by plants as conserved features from potential enemies. Strikingly, PCs can also be found in about 10% of bacterial species, including bacteria engaged in microbe-host interactions (*Aktas et al., 2010*). Besides insect eggs, PCs may thus also constitute MAMPs from bacterial or fungal pathogens. How and if plants discriminate PCs originating from insect eggs and bacterial pathogens is an intriguing question that deserves future investigation.

In summary, we report here the identification of PCs as EAMPs from insect eggs. PCs represent a class of conserved molecules that have the ability to trigger immune defenses in *Arabidopsis*. This study illustrates the sophistication of plant-herbivore interactions and expands the repertoire of patterns that plants use to recognize insect attack.

# Materials and methods

## Key resources table

| Reagent type (species) or resource | Designation | Source or reference | Identifiers | Additional information |
|---|---|---|---|---|
| Gene (*Arabidopsis thaliana*) | PR1 | arabidopsis.org | At2G14610 | |
| Gene (*Arabidopsis thaliana*) | SAG13 | arabidopsis.org | At2G29350 | |
| Gene (*Arabidopsis thaliana*) | TI | arabidopsis.org | At1g73260 | |
| Gene (*Arabidopsis thaliana*) | LecRK-I.8 | arabidopsis.org | At5g60280 | |
| Genetic reagent (*Arabidopsis thaliana*) | *lecrk-I.8* T-DNA | Nottingham Arabidopsis stock center (NASC) | SALK_066416 | |
| Genetic reagent (*Arabidopsis thaliana*) | PR1::GUS | *Bruessow et al., 2010* | | |
| Genetic reagent (*Arabidopsis thaliana*) | SAG13::GUS | *Bruessow et al., 2010* | | |
| Genetic reagent (*Arabidopsis thaliana*) | TI::GUS | *Bruessow et al., 2010* | | |
| Genetic reagent (*Arabidopsis thaliana*) | *sid2-1* | *Nawrath and Métraux, 1999* | | |
| Genetic reagent (*Acinetobacter sp. ADPWH_lux.*) | bacterial biosensor | *Huang et al., 2005* | | |
| Sequence-based reagent | PR1_FWD | This paper | PCR primers | GTGGGTTAGCGAGAAGGCTA |
| Sequence-based reagent | PR1_RV | This paper | PCR primers | ACTTTGGCACATCCGAGTCT |
| Sequence-based reagent | SAG13_FWD | This paper | PCR primers | GTCGTGCATGTCAATGTTGG |
| Sequence-based reagent | SAG13_RV | This paper | PCR primers | CCAAGGACAAACAGAGTTCG |
| Sequence-based reagent | TI_FWD | This paper | PCR primers | CCTCGTGGTTGCTGGTCCAAA |
| Sequence-based reagent | TI_RV | This paper | PCR primers | CCTCTCACATAGTCTTGGACGAAA |
| Sequence-based reagent | SAND_F | This paper | PCR primers | AACTCTATGCAGCATTTGATCCACT |
| Sequence-based reagent | SAND_R | This paper | PCR primers | TGATTGCATATCTTTATCGCCATC |

## Plant material and insect growth conditions

*Arabidopsis thaliana* plants were vernalized for 2 days at 4°C and cultivated in pots containing moist compost. Plants were grown in a controlled environmental chamber with a 10-hr day / 14-hr night cycle as described previously (*Reymond et al., 2004*). Experiments were conducted with four- to five-week-old plants. The *lecrk-I.8* T-DNA (SALK_066416) (*Gouhier-Darimont et al., 2013*) and *sid2-1* (*Nawrath and Métraux, 1999*) mutants, and the GUS reporter lines PR1::GUS, SAG13::GUS, and TI::GUS (*Bruessow et al., 2010*) were described previously.

A colony of *P. brassicae* (large white butterfly) was reared on *Brassica oleracea* var. *gemmifera* as described previously (*Bonnet et al., 2017*). *Spodoptera littoralis* eggs were obtained from Syngenta (Dr. O. Kindler, Stein, Switzerland).

## EE preparation and purification of defense eliciting *P. brassicae* egg lipids

EE was prepared from *P. brassicae* eggs collected from cabbage leaves. *P. brassicae* eggs were crushed with a pestle in a 1.5 mL reaction tubes and centrifuged for 3 min at 15'000 g. The supernatant (EE) was stored at −20°C. Natural oviposition and EE application to plants have been described previously (*Gouhier-Darimont et al., 2013*; *Gouhier-Darimont et al., 2019*). After three days, *P. brassicae* eggs and the EE were carefully removed from treated leaves before sample analysis.

Egg lipid purification with Cleanascite was conducted following the manufacturer's instructions (Biotech Support Group LLC). Briefly, EE was mixed with Cleanascite 1:1 (V:V) by gently shaking for 10 min and centrifuged for 15 min at 10'000 g for phase separation. For application of lipid-free supernatant (CS SN) and lipid fraction (CS LF), 2 μL of CS SN and CS LF were spotted under two leaves of each treated plant. Plants were treated for three days. Untreated plants and plants treated with EE diluted 1:1 with $H_2O$ served as controls. Three plants were used for each treatment.

For extraction and purification of eggs lipids, ca. 100'000 *P. brassicae* eggs were collected over the course of several weeks, yielding 10 mL of EE. Aliquots of 1 mL of *P. brassicae* EE were then mixed dropwise with 6.25 mL of $CHCl_3$:EtOH (1:1; V:V) and mixed on a shaker for 1 hr at room temperature with additional 15 mL of $CHCl_3$:EtOH. The extract was dried under a nitrogen stream and resuspended in 25 mL $CHCl_3$. After filtration through a funnel packed with cotton, $CHCl_3$ was evaporated and the lipids (lipid fraction, LF) resuspended in 10% DMSO. LFs from 10 × 1 mL aliquots (corresponding to 1.1 g total lipids) were pooled and further separated by SPE. For plant treatment, LF was solved at 5 μg/μL in 1% DMSO by sonication.

For SPE fractionation, 1.1 g of LF was mixed with 7 g of silica gel and loaded to a *ZEO*prep 60 C18 reverse-phase cartridge (40–63 μm; BGB Analytics AG, Boeckten, Switzerland) and sealed with sand. The cartridge was placed in a PuriFlash 400 system and eluted with 300 mL of 25% MeOH (Fr. 1), followed by 300 mL of 50% MeOH (Fr. 2), 350 mL of 75% MeOH (Fr. 3), 550 mL of 100% MeOH (Fr. 4) and 250 mL of 100% $C_4H_8O_2$ (Fr. 5). SPE fractions were dried under a nitrogen stream. For plant treatment, aliquots of SPE fractions were solubilized in 1% DMSO at 5 μg/μL by sonication. Eight leaves from four plants (two leaves per plant) were treated with a 2 μLdrop of each SPE fraction from the abaxial side of the leaf and treated leaves were harvested three days later for analysis. Plants treated with LF and 1% DMSO served as controls.

Further fractionation of Fr. four was performed on a semi-preparative HPLC equipment (Armen modular spot prep II, Saint-Avé, France) connected to an ELSD Sedex LT-ELSD 85 (Sedere, Alfortville, France). The fractionation was performed on 30 mg using a reverse-phase semi-preparative X-bridge C18 column (150 × 19 mm, 5 μm; Waters, Milford, MA, USA), with water (A) and methanol (B) containing both 0.1% formic acid as mobile phase. Separation was performed with a step gradient from 5% to 96% of B in 60 min, held during 30 min, then 96% to 100% of B in 20 min, held during 20 min. The flow rate was fixed at 17 mL/min. The UV detection in the scan mode (210–366 nm) and the ELSD conditions at 1 mL/min, 40°C, 3.1 bar $N_2$ and gain 8. Separation of Fr. four led to 17 subfractions (Fr. 4.1 to Fr. 4.17) with clearly distinguishable ELSD signal, which were eluted during the isocratic phase of the gradient (96% of B, 60–90 min) and were collected, evaporated under a nitrogen stream and used for further treatments and analyses. For plant treatment, aliquots of HPLC fractions were solubilized in 1% DMSO at 5 μg/μL by sonication. Eight leaves from four plants (two leaves per plant) were treated with a 2 μL drop of each HPLC fraction from the abaxial side of the leaf and treated leaves were harvested three days later for analysis. Plants treated with SPE Fr. four (5μg/μL) served as controls.

## Detection of egg lipids on filter paper

Filter paper sheets were clipped on leaves from *B. oleracea* and placed in a cage with adult *P. brassicae* butterflies. Sheets containing egg batches were removed and left at room temperature for one day. Then, eggs were gently removed and the oviposited area was cut and 2–4 filter paper pieces (equivalent to 80 eggs) were placed into a 1.5 mL glass vial containing 600 μL MeOH. After gentle

agitation for 10 min, the pieces were removed and the solution was dried with $N_2$ before further analysis. Filter paper sheets clipped on *B. oleraceae* leaves but not exposed to butterflies were used as controls. For *S. littoralis*, egg batches were obtained on filter paper three days after oviposition. Pieces of 0.5 cm² (150–200 eggs) were cut and placed in MeOH.

## Nuclear magnetic resonance (NMR) spectroscopy

The samples were dissolved in $CD_3OD$ and the NMR experiments were recorded on a Bruker Avance III HD 600 MHz NMR spectrometer equipped with a QCI 5 mm Cryoprobe and a SampleJet automated sample changer (Bruker BioSpin, Rheinstetten, Germany). For assignment of $^{31}P$ signals the heteronuclear TOCSY pulse sequence of *Kellogg, 1992* modified by *Balsgart et al., 2016* was employed with a mixing time of 70 ms. The 2D $^1H-^{31}P$ NMR experiments were recorded using 256 t1 increments each consisting of 32 scans with a repetition delay of 1 s and eight dummy scans. The quantitative 1D $^{31}P$ NMR experiments were recorded under inverse gated decoupling to avoid NOE transfer from $^1H$ to $^{31}P$ and employed 128 scans and repetition delays of 20 s. Chemical shifts are reported in parts per million (δ) using the residual $CD_3OD$ signal ($\delta_H$ 3.31) as internal standard for $^1H$ and the TIBP (triisobutyl phosphate) signal ($\delta_P$ −0.36) as internal standards for $^{31}P$ NMR. The PC quantification was performed either by the ERETIC method (*Akoka et al., 1999*) using the triphenyl phosphate solution as external reference (48.5 mM) or by adding TIBP as internal standard (1.5 mM). All NMR spectra were recorded at 298 K.

## MS analysis of lipids in HPLC fractions and on filter paper

Dried fractions were dissolved in chloroform/methanol (1:1 v/v) and diluted into chloroform/methanol (1:2) containing 5 mM ammonium acetate, which were then infused into a Triple Stage Quadrupole Vantage (Thermo Scientific) equipped with a Triversa NanoMate (Advion) and analyzed by full scan in positive and negative modes as well as precursor ion scans for PCs (positive mode, *m/z* 184.074, collision energy 30). The data were analyzed using Xcalibur (Thermo Scientific). Relatively little material was detected in the negative ion mode and in the positive mode, the major species were found in the precursor ion scan for PCs. Possible PC species were identified by their masses, giving a PC with x number of C and y number of double bonds (eg. m/z 732.4 = PC(32:1)). To identify fatty acyl chains, LiAc was added to 5 mM and fragmentation of the desired PC species was performed using the compound optimization method for fragmentation (TSQ Tune) (*Hsu and Turk, 1999*). Loss of specific fatty acids were detected by neutral ion loss analysis.

For samples on filter paper, dried samples were resuspended in 100 microliters of chloroform/methanol/water (2:7:1, with 5 mM ammonium acetate) and infused into the TSQ triple quadrupole MS. A full scan was performed in the positive mode to see all positively charged lipids. A calibration curve was done with 0.05, 0.5 and 5 μg PC(16:1/16:1) spotted on filter paper and extracted like egg samples.

## MS shotgun lipidomic analysis

MS analysis of *P. brassicae* and *S. littoralis* EE was performed at Lipotype GmbH (Dresden, Germany) according to *Surma et al., 2015*. Lipid profiling was conducted in triplicates for both insect species.

## Phospholipase treatment of EE

For phospholipase treatment, 100 μL of EE were mixed with 20 U of Phospholipase D (Sigma-Aldrich, P8398) (PLD) and 20 U of Phospholipase $A_2$ (Sigma-Aldrich, P6534) ($PLA_2$) and incubated for 1.5 hr at 30℃ followed by incubation at 37℃ for another 1.5 hr. For separate phospholipase treatments, 100 μL of EE were mixed with 20 U PLD or PLA2 and incubated for 1.5 hr at 30℃ for PLD or 1.5 hr at 37℃ for $PLA_2$. Eight leaves from four plants (two leaves per plant) were treated from the abaxial side of the leaf with a 2 μL drop of the phospholipase-treated EE or an untreated EE which was incubated in the same way as described above. Treated leaves were harvested three days later for analysis and untreated plants served as controls.

Efficacy of PC degradation in phospholipase-treated EE was checked by $^{31}P$ 1D NMR (see above).

## Phospholipid treatments and wounding

Pure phospholipids for plant treatments were ordered from Avanti Polar Lipids (Alabaster, Alabama, USA) PC(3:0/3:0), 1,2-dipropionyl-sn-glycero-3-phosphocholine, 850305; PC(16:0/16:0), 1,2-dipalmitoyl-sn-glycero-3-phosphocholine, 850355; PC(16:0-18:1), 1-palmitoyl-2-oleoyl-glycero-3-phosphocholine, 850457; PC(16:1/16:1), 1,2-dipalmitoleoyl-sn-glycero-3-phosphocholine, 850358; PC(18:0-18:1), 1-stearoyl-2-oleoyl-sn-glycero-3-phosphocholine, 850467; PC(18:1/18:1), 1,2-dioleoyl-sn-glycero-3-phosphocholine, 850357; PC(18:3/18:3), 1,2-dilinolenoyl-sn-glycero-3-phosphocholine, 850395; PE(18:1/18:1), 1,2-dioleoyl-sn-glycero-3-phosphoethanolamine, 850725; PA(18:1/18:1), 1,2-dioleoyl-sn-glycero-3-phosphate, 840875; DAG(16:0/18:1), 1-palmitoyl-2-oleoyl-sn-glycerol, 800815; TAG(18:1/18:1/18:1), 1,2,3-tri-(9Z-octadecenoyl)-glycerol, 870110; LPC(18:1), 1-hydroxy-2-oleoyl-sn-glycero-3-phosphocholine, 845875; LPE(18:1), 1-oleoyl-2-hydroxy-sn-glycero-3-phosphoethanolamine, 846725; PC-Mix purified from chicken egg, 840051; Lyso-PC-Mix purified from chicken egg, 830071; PE-Mix purified from chicken egg, 840021; Lyso-PE-Mix purified from chicken egg, 860081. PA-Mix purified from chicken egg, 84010; SM-Mix purified from chicken egg, 860061. Phospholipid stock solutions were made in 100% MeOH analytical grade. For phospholipid application, an appropriated amount of the stock solution was transferred in a fresh tube, the MeOH was evaporated under a nitrogen-flux and the phospholipids were solved in 1% DMSO, 0.5% Glycerol and 0.1% Tween 20 by sonication. For GUS staining analysis, six leaves of two plants (three leaves per plant) were treated, for qPCR analysis eight leaves of four plants (two leaves per plant) were treated, and for DAB and trypan blue staining nine leaves of three plants (three leaves per plant) were treated. Plants were treated from the abaxial side of the leaf with a 2 μL drop of the phospholipid solution and harvested after three days. Control plants were treated with 1% DMSO, 0.5% Glycerol and 0.1% Tween 20. Oleic acid (C18:1) (Sigma-Aldrich, O1008) solution was prepared as described for phospholipids. Choline (Sigma-Aldrich, C7017) and phosphocholine (Sigma-Aldrich, P0378) stock solutions were made in H₂O and diluted in the control solution for plant treatments. The concentrations used for each separated or combined phospholipid or chemical treatment are given in the corresponding figure legends.

For wounding experiment, a one-side corrugated forceps wounded the leaves mostly from the abaxial site. The size of the corrugated wounding-surface was adjusted to the size of a 2 μL drop of EE. Samples were harvested after three days.

## RNA sequencing experiment

For experiments with natural oviposition, 15 plants were placed in a 60 × 60×60 cm tent containing approximately 30 *P. brassicae* butterflies. After 24 hr, four plants containing one egg batch on two leaves were placed in a growth chamber for four days. Just before hatching, eggs were gently removed with a forceps. For EE application, 2 × 2 μL of EE were spotted under the surface of each of two leaves of each treated plant. Four plants were treated with EE for 5 days. Treated or oviposited leaves were harvested and quickly stored in liquid N₂. Untreated plants were used as controls.

Total RNA from three biologically independent experiments was extracted using an RNeasy plant mini kit (Qiagen). DNase treatment was added to the protocol. For cDNA synthesis, RNA samples were purified by NaAC 3M and EtOH precipitation. Library were synthetized from 500 ng of purified total RNA using the TrueSeq stranded mRNA kit (Illumina). RNA and library quality was assessed with a fragment analyzer from Advanced Analytical. Library were sequenced with the Illumina HiSeq 2500 sequencer at the Genomic Technologies Facility platform of the University of Lausanne (LGTF) (http://www.unil.ch/gtf/en/home.html). Libraries were multiplexed and sequenced twice aiming to obtain 35 million reads per sample. Reads were mapped to the *Arabidopsis* TAIR10 genome release using STAR (*Dobin et al., 2013*). Counts were normalized according to the Trimmed Mean of M-values (TMM) method and per library size. Data were transposed in normalized counts/million reads (in log2) to be further analyzed using R Bioconductor package LIMMA. Data have been deposited in the National Center for Biotechnology Information's Gene Expression Omnibus (GEO) under the accession number GSE144091.

## Gene expression analysis

Total RNA was extracted using the Relia Prep RNA Tissue Mini Prep System (Promega) following the manufacturer's instructions. cDNA synthesis was conducted using 1 μg of total RNA for reverse-

transcription by M-MLV reverse transcriptase (Invitrogen) in a final volume of 25 µL. cDNA was synthesized in triplicates and diluted eightfold with water for subsequent quantitative real-time PCR (qPCR) analysis. qPCR analysis was performed with gene specific primers for *PR1* (At2g14610; FW: 5′-GTGGGTTAGCGAGAAGGCTA-3′, RV: 5′-ACTTTGGCACATCCGAGTCT-3′), *SAG13* (At2g29350; FW: 5′-GTCGTGCATGTCAATGTTGG-3′, RV: 5′-CCAAGGACAAACAGAGTTCG-3′), and *TI* (At1g73260; FW: 5′-CCTCGTGGTTGCTGGTCCAAA-3′, RV: 5′-CCTCTCACATAGTCTTGGACGAAA-3′) in a final volume of 20 µL containing 2 µL of cDNA, 0.2 µM of each primer, 0.03 µM of reference dye (ROX), and 10 µL of Brilliant III Ultra Fast SYBR Green QPCR Master Mix (Agilent) on a QuantStudio three real-time PCR machine (Applied Biosystems; Thermo Scientific) with the following program: 95°C for 3 min, then 40 cycles of 10 s at 95°C and 20 s at 60°C. mRNA levels were normalized to the house keeping gene *SAND* (At2g28390; FW: 5′-AACTCTATGCAGCATTTGATCCACT-3′, RV: 5′-TGATTGCATATCTTTATCGCCATC-3′).

## Histochemical staining

GUS staining and visualization of cell death by trypan blue staining were done as described previously (*Little et al., 2007*). Detection of hydrogen peroxide by 3,3′-diaminobenzidine (DAB) staining was conducted as described previously (*Daudi and O'Brien, 2012*).

## Measurement of total SA

Determination of total SA was done using the bacterial biosensor *Acinetobacter* sp. ADPWH_lux. (*Huang et al., 2005*; *Huang et al., 2006*) according to *Defraia et al., 2008* with minor modification. Briefly, 18 leaves of six plants (three leaves per plant) were exposed to natural oviposition, treated with 2 µL of EE, PC-Mix, PC(16:1/16:1) or PC(18:3/18:3), or wounded. Untreated plants and plants treated with the solvent control served as controls. Three days later, six leaf discs of 0.7 cm diameter from two plants were harvested per sample, combined and the fresh weight was determined. Samples were ground in liquid nitrogen and extracted in 200 mL of 0.1 M sodium acetate buffer (pH 5.6) and centrifuged at 4°C for 15 min at 16'000 g. 50 µLof the extract was incubated with 5 µL β-glucosidase from almonds (Sigma-Aldrich, G0395; 0.5 U/µL in acetate buffer) for SA release from SA-glucoside for 90 min at 37°C. Afterwards, 20 µL of the extract was mixed with 60 µL of Luria Broth medium and 50 µL of an overnight log phase culture of *Acinetobacter* sp. ADPWH_lux (OD600 = 0.4), and incubated for 1 hr at 37°C. Luminescence was measured with a Hidex microplate reader using a 485 ± 10 nm filter for 1 s. For absolute SA quantification, a SA standard curve (0–60 ng) in untreated *sid2-1* leaf extracts was read in parallel.

## Statistical analyses

We assume normal distribution of values for gene expression levels. Because of the large range of relative expression values all statistical tests on gene expression data were performed on log-transformed data. Determination of statistically significant differences between gene expression samples was evaluated by analysis of variance (ANOVA) followed by Tukey's honest significant. Because criteria for homogeneity of variance were not always met (Levene test), statistical differences for pairwise comparisons were evaluated by a two-sided Welch's $t$-test. When values were not normally distributed (Shapiro-Wilk test), we used the Mann-Whitney $U$ test. All statistical tests were done in R (RStudio version 1.1.442). The choice of statistical test is given in the corresponding figure legend.

## Acknowledgements

We thank Blaise Tissot for help in growing plants and the Centre for Ecology and Hydrology from the Natural Environment Research Council (Swindon, UK) for providing *Acetinobacter* sp. ADPWH_lux. We are grateful to Guillaume Marti for MS analysis in the preliminary phase of the project. We thank Niko Geldner for critically reading the manuscript and Emanuel Schmid-Siegert for help in analyzing RNAseq data. The Swiss SNF (grant 31003A_169278 to P R and grant 166686 to HR) supported this work as well as the Swiss National Centre for Competence in Research in Chemical Biology. The ISPWS (Prof. J-L Wolfender) is thankful to the Swiss SNF for the support in the acquisition of the NMR 600 MHz (SNF R'Equip grant 316030_164095).

# Additional information

### Funding

| Funder | Grant reference number | Author |
|---|---|---|
| Schweizerischer Nationalfonds zur Förderung der Wissenschaftlichen Forschung | 31003A_169278 | Philippe Reymond |
| Schweizerischer Nationalfonds zur Förderung der Wissenschaftlichen Forschung | 166686 | Howard Riezman |
| Schweizerischer Nationalfonds zur Förderung der Wissenschaftlichen Forschung | 316030_164095 | Jean-Luc Wolfender |

The funders had no role in study design, data collection and interpretation, or the decision to submit the work for publication.

### Author contributions

Elia Stahl, Conceptualization, Investigation, Writing - review and editing; Théo Brillatz, Emerson Ferreira Queiroz, Laurence Marcourt, Investigation, Methodology, Writing - review and editing; André Schmiesing, Olivier Hilfiker, Isabelle Riezman, Investigation, Writing - review and editing; Howard Riezman, Jean-Luc Wolfender, Formal analysis, Investigation, Methodology, Writing - review and editing; Philippe Reymond, Conceptualization, Supervision, Validation, Writing - original draft, Project administration, Writing - review and editing

### Author ORCIDs

Emerson Ferreira Queiroz (iD) http://orcid.org/0000-0001-9567-1664
Howard Riezman (iD) http://orcid.org/0000-0003-4680-9422
Jean-Luc Wolfender (iD) http://orcid.org/0000-0002-0125-952X
Philippe Reymond (iD) https://orcid.org/0000-0002-3341-6200

### Decision letter and Author response

Decision letter https://doi.org/10.7554/eLife.60293.sa1
Author response https://doi.org/10.7554/eLife.60293.sa2

# Additional files

### Supplementary files

• Supplementary file 1. Arabidopsis transcriptome after 5 days of *P. brassicae* oviposition or treatment with egg extract. Plants were naturally oviposited by *P. brassicae* butterflies and leaves containing one egg batch (ca. 20-30 eggs) were harvested after 5 days. For EE treatment, leaves were treated with 2 x 2 microL of EE (corresponding to ca. 30 eggs) and harvested after 5 days.

• Transparent reporting form

### Data availability

RNAseq data have been deposited in GEO under accession code GSE144091.

The following dataset was generated:

| Author(s) | Year | Dataset title | Dataset URL | Database and Identifier |
|---|---|---|---|---|
| Hilfiker O, Reymond P | 2020 | Conserved Arabidopsis response to oviposition and crude egg extract treatment | https://www.ncbi.nlm.nih.gov/geo/query/acc.cgi?acc=GSE144091 | NCBI Gene Expression Omnibus, GSE144091 |

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
