## [Decision Letter]

**Decision letter after peer review:**

[Editors’ note: the authors submitted for reconsideration following the decision after peer review. What follows is the decision letter after the first round of review.]

Thank you for submitting your work entitled "Phosphatidylcholines from insect eggs activate defense responses in *Arabidopsis*" for consideration by *eLife*. Your article has been reviewed by three peer reviewers, including Meredith Schuman as the Reviewing Editor and Reviewer #1, and the evaluation has been overseen by a Reviewing Editor and a Senior Editor.

This study is interesting and may represent a significant advance in our understanding of plant responses to insect oviposition. In particular, the use of appropriate bioassays, a broad set of response markers, egg extracts rather than insect extracts, sufficient structural elucidation, and transgenic lines to investigate possible plant response mediators, are all strengths.

However, the reviewers expressed three sets of concerns which must be resolved before it is clear how the data are to be interpreted, and specifically whether the claim in the title, that phosphatidylcholines from insect eggs activate defense responses in *Arabidopsis*, can be supported (see also comment 5). These are detailed below.

Thus we have decided to decline the manuscript in its current form, because we do not know whether these concerns can be addressed sufficiently within the usual time we allow for revisions at *eLife*, and because to do so may require some additional experimental work. However, should you be able to address these concerns, we would encourage you to submit a revised manuscript to *eLife* again as a new submission.

Essential revisions:

1) The authors have good evidence that egg-derived phospholipids largely explain the transcriptional response to oviposition and at least two associated stress responses (SA accumulation, cell death) in the *P. brassicae*-*A. thaliana* interaction, but the evidence for the specific importance of egg-derived PCs is weaker. Specifically:

1.1) PCs clearly do not account for the entire egg lipid-associated response, which the authors acknowledge and discuss.

1.2) Tests with purified PCs and PEs use chicken eggs as a source and the composition is not compared to the composition of PCs and PEs found in *P. brassicae* eggs.

1.3) PLA2 also cleaves other phospholipids, as does PLD (albeit with weaker activity), and thus the phospholipase assay demonstrates the importance of phospholipids but not PCs specifically; Figure 2F is misleading and the authors should show the full lipid spectrum.

2) The authors need to test whether leaf-derived PCs (and perhaps other phospholipids, see 1) could also explain the response they observe, in which case the claim that these are "EAMPs" cannot be supported and this rather starts to look like a less specific damage-associated response.

3) The authors show that there are sufficient concentrations of PCs in eggs to elicit a response, but do not show that oviposited eggs actually release such concentrations to the leaf. This could in part be addressed by comparing the response of WT plants and LecRK-I.8 mutants to oviposition as well as to egg extract and an appropriate PC or phospholipid mix (again, see point 1).

4) Reviewer 1 requests that the authors document their statistical approach in the Materials and methods section (software used, testing that assumptions of statistical tests were met) and that the authors be ready to submit source data for all figures upon publication.

5) Reviewer 2 points out that the elicited response in *Arabidopsis* has not been shown to defend against *P. brassicae*, and thus the authors should not refer to this as a defense response but might rather characterize it as an immune response.

6) Reviewer 3 notes that phosphatidic acid (PA) was not tested, while the authors' own reasoning (shown in Figure 3B) indicates that this is relevant.

We have included the individual peer reviews here for your information. However, please note that the comments above reflect the consensus of the reviewers and their common recommendation.

Reviewer #1:

Phosphatidylcholines are not novel oviposition-associated elicitors, but had not previously been shown to be elicitors originating from oviposited eggs. A phosphatidylcholine and three other phospholipids were previously identified by Yang and colleagues from extracts of *Sogatella furcifera* insects, https://doi.org/10.1080/09168451.2014.917266. Stahl and colleagues refer to this work in their Discussion section. In this earlier work, Yang and colleagues used bioassay-guided fractionation of female adult extracts eliciting benzyl benzoate accumulation in Oryza sativa L., and identified four similar compounds present in hundreds of ng to ug amounts per extracted female. The structure of one of these compounds was determined by ^1^H-NMR, ^13^C-NMR, methanolysis and GC-MS analysis and comparison with the GC-MS analysis of a standard substance, to be 1,2-dilinoleoyl-sn-glycero-3-phosphocholine. The authors then demonstrated that pure 1,2-dilinoleoyl-sn-glycero-3-phosphocholine in a similar concentration also elicited benzyl benzoate accumulation in rice.

The current work by Stahl and colleagues goes beyond the study by Yang and colleagues in several respects, and as a result the work of Stahl and colleagues represents a significant advance in the elucidation of egg-associated elicitors:

- Stahl and colleagues demonstrate that phospholipids from egg extracts, rather than insect extracts, elicit a plant response.

- They use multiple oviposition-associated plant responses to assess the similarity of plant responses to phosphatidylcholines, versus egg extract, rather than only a single response.

- They demonstrate that LecRK-I.8 is involved in eliciting the phosphatidylcholine-mediated plant response to egg extract, in that *Arabidopsis thaliana* knock-outs of LecRK I.8 show a substantially attenuated response to both phosphatidylcholines (mix and pure) as well as egg extract.

Stahl and colleagues furthermore present additional evidence that it is the lipidic fraction of egg extract which is most active, and cleverly show that elimination of phospholipids using phospholipase removes fraction activity.

Prior to the invitation of this manuscript for full review, I had requested that the authors better justify their choice of marker genes and also demonstrate to the best of their ability that egg extract, which is their positive control for most of this study, elicits a response substantially similar to oviposition. They have done so, in new supplements to Figure 1 showing results from an RNA-seq experiment comparing the transcriptomic response of *A. thaliana* plants to egg extract, versus oviposition.

In summary, the work by Stahl and colleagues could represent a substantial advance in the elucidation of egg-associated elicitors and the responses they trigger in plants.

The title is well chosen if the claims can be fully supported, although the authors might consider adding the insect species name. The manuscript is well and clearly written and the figures are clearly presented. It is easy to follow the authors' experimental logic. To my knowledge, they appropriately refer to and discuss relevant literature.

The authors, however, do not describe their statistical analyses in their methods section, and this needs to be rectified. See comments below. I would also encourage them to submit all source data.

Reviewer #2:

The intriguing study by Stahl et al., aims to identify the molecular patterns of insect eggs that activate an immune response in *Arabidopsis thaliana*. By performing NMR, LCMS spectrometry and lipidomic analysis as well as bioactivity-guided fractionation using expression of three reporter/defence genes, the authors concluded that phosphatidylcholines (PCs), a class of phospholipids common components of cell membranes, are present in the active fractions and act as egg-associated molecular patterns (EAMPs). Although we find the study highly exciting and mostly elegantly performed, we have a few major concerns, namely:

a) whether the PCs were the only active compounds in egg extract lipid fraction;

b) that the PCs shown to be bioactive were actually not purified from the butterfly eggs (as claimed in the title and in the Discussion section);

c) that plant-derived PCs have not been tested in order to rule out whether they are likely to induce a similar response (we would need to call them then DAMPs (damage-associated molecular patterns) instead of EAMPs);

d) whether the bioactivity of PC (16:1/16:1) is dependent on the whole molecule since lisophosphatidylcholine (released with a fatty acid by PLA2-mediated PC cleavage) and phosphatic acid (released with a choline by PLD-mediated cleavage) have been previously shown to be bioactive in plants and;

e) that the study might not be novel enough (as claimed in the abstract) as similar EAMPs were found in planthopper eggs.

Not all lipids present in the *PR1*-inducing subfractions 4.10-4.17 were tested, like SMs (sphingomyelins) and PEs (phosphatidylcholines). These two appear to be present in most of the *PR1*-active fractions but were not further evaluated for their activity as pure compounds (except the PE mix from chicken eggs). The authors motivate the focus on PCs based on the PLD + PLA2-treated egg extract that lose *PR1* induction (Figure 2E). PLD and PLA2 could be active also on PEs, thus PEs should still be tested as candidates. Furthermore, Figure 2F should also show the effect of PLD + PLA2 across the whole lipid NMR spectra not only on PCs. The ultimate proof that insect-derived PCs are active, should be done by testing PCs purified form the HPLC bioactive subfractions 4.10-4.17. Assuming the activity of insect egg derived PCs, the questions on their origin is still unresolved, given their ubiquity in plant cells.

Lastly, the authors claim that their results point to the conservation of phospholipid EAMPs between distantly related insect species, including moths, planthoppers or true flies. However, most eukaryotes incl. plant membranes and some bacteria contain PCs, and, in our view, it cannot be stated yet that this immunity response is specific to insect eggs. And that brings us to our last major concern that *A. thaliana* is unlikely to have evolved such a response to defend insect eggs: neither is it a host plant for one of the tested lepidopteran species nor has it been shown to actually defend eggs, by e.g. impeding egg survival.

Reviewer #3:

The manuscript by Stahl et al., is a continuation of published work which showed that deposition of insect eggs on *Arabidopsis* leaves induces a set of defense responses known from treatments with MAMPs from microbial pathogens and DAMPs released by wounding. The authors previously also showed that induction of these responses can be mimicked by applying egg extracts (EE) and that responsiveness to EE is reduced but not abolished in mutants lacking the receptor like kinase LecRK-I.8. In the present manuscript the authors provide a comprehensive and convincing set of data that identify the active factor in EE as phosphatidylcholine (PC). Thereby, as observed for the response to crude EE, the response to PC was significantly reduced in mutants lacking LecRK-I.8. Based on these results the authors conclude that oviposition is detected via direct or indirect detection of PC by LecRK-I.8.

However, as a major concern from the side of this reviewer, this conclusion seems not justified since the authors provide no direct evidence that the response to oviposition indeed proceeds via (substantial) release of PC from the intact insect eggs on the leaf surface. A first line of evidence supporting such a process could consist in experiments demonstrating that the response to oviposition is equally dependent on LecRK-I.8 as the response to PC and EE. Such evidence is missing in the current work and seems to be missing also in the previous publications on LecRK-I.8 (at least I could not detect such data on first glance in the publications available to me).

An experimentally confirmed activation mechanism with *Arabidopsis* reacting to oviposition via (direct or indirect) detection of such a common molecule as PC would certainly add a new and intriguing aspect as to how plants can detect adverse changes in their environment.

[Editors’ note: further revisions were suggested prior to acceptance, as described below.]

Thank you for resubmitting your work entitled "Phosphatidylcholines from *Pieris brassicae* eggs activate an immune response in *Arabidopsis*" for further consideration by *eLife*. Your revised article has been evaluated by Meredith Schuman (Senior Editor) who also served as Reviewing Editor.

The reviewers agree that the revisions have fully addressed the concerns expressed in the prior round of review and unanimously recommend acceptance; however, three revisions are required before the article is formally:

1) Authors should briefly explain the two different nomenclatures used to refer to phospholipids in order to avoid reader confusion, as I expect that readers with a background in molecular biology or plant-insect interactions may not immediately realize the parity (e.g. 36:2 versus 18:1/18:1). This can be quickly done in the explanations of Figure 2 and Figure 3.

2) Variation in egg-induced necrosis in *A. thaliana* has been reported suggesting "some ecological relevance in the model species" (Introduction).It may still be too early to draw such a conclusion in absence of reports on *A. thaliana* expressing a necrosis sufficient to reduce egg survival, considering also that *A. thaliana* has not yet been reported as a natural host for *P. brassicae*. Please revise wording.

3) The authors should note whether and how assumptions of their chosen statistical tests were evaluated (homogeneity of variance, normal distribution) and whether data were transformed prior to analysis if these assumptions were not met.

The above also constitutes my review of the revised manuscript. Below, for your information, are the comments of the other reviewers. However, please note that the assessment above is the agreed-upon recommendation, and no additional responses to individual reviews are necessary.

Reviewer #2:

I´m fully satisfied by the responses and revisions provided by the authors. In my opinion, this very careful and thorough study, demonstrating that also a very common metabolite can mediate a defense responses in plants, will make a nice and valuable contribution to *eLife*.

Reviewer #3:

The revised version of this manuscript by Stahl et al. provides convincing evidences in support of a role for phosphatidylcholines (PC) as insect egg elicitors of a plant immune response. The authors succeeded in addressing timely and elegantly all the concerns that we expressed in our first review with few additional experiments.

Bioassays using a synthetic egg liquid mix (SELM) confirm the link between the bioactivity of the egg extract and of PCs, ruling out a possible effect of the other components of the egg lipid fraction (LPCs, LPEs, PEs and SMs). The authors acknowledge the impossibility to fully purify PCs from insect eggs given the current state of the art. Nonetheless, they provide data that show a very similar fatty acid composition between the PCs from the *P. brassicae* eggs and the PCs derived from chicken eggs that were deployed in the bioassays.

Further, the authors provide also a more complete picture on the specificity of the PCs components that are produced by phospholipases cleavage. Besides the whole PC molecule, only phosphatidic acid (PA) shows an induction of the plant immune response, which is in accordance to previous studies. Interestingly, PA activity appears somehow lower than the whole PC, perhaps suggesting that the specificity of the elicitor bioactivity resides in other components/epitopes.

Wounding treatment was added to the different bioassays without showing any induction of the plant immune response, thus ruling out also a possible plant origin for the bioactive PCs.

The total PCs released by an egg clutch (~20 eggs) after one day were quantified in 0.23 µg. This amount is about 50x lower than the 5 µg of total PCs that were found on an EE of 20 eggs. Unlike what is stated by the authors (Discussion section), given the current data it is not yet possible to assume a daily constant rate of diffusion of 0.23 µg PCs from the egg into the plant. Thus, the use of 1 µg/µl of PCs (or derivatives) may be still higher than the amount that are present on the leaf surface under an egg clutch. Nevertheless, the authors show that PCs at as low concentration as 0.02 µg/µl are enough to induce a response, albeit weaker than the egg extract.

The authors correctly indicate that PCs are classes of molecules shared also by bacterial and fungal kingdoms, thus opening the intriguing question how and if a plant can discriminate PCs from different attacker organisms. Hopefully future studies will shade light on these questions.

[Editors’ note: further revisions were suggested prior to acceptance, as described below.]

Thank you for resubmitting your work entitled "Phosphatidylcholines from *Pieris brassicae* eggs activate an immune response in *Arabidopsis*" for further consideration by *eLife*. Your revised article has been evaluated by Meredith Schuman (Senior Editor) and a Reviewing Editor.

We are ready to accept your manuscript conditional on a few additional changes which, however, should be fast to make.

The last round of review requested that the authors state how they determined the suitability of statistical tests (e.g. homogeneity of variance, normal distribution) and whether any data were transformed. The authors have added corresponding statements to the text, but these do not always seem to fit with the data as shown in the figures. In general, it looks as though it would be necessary to either log-transform or use a nonparametric test for most of the quantitative transcript accumulation data due to the very large differences in variance as a result of the control levels being near the limit of detection (e.g. Figure 1B, Figure 2E and G). It is also strange that the same problems with homogeneity of variance were not encountered in both Figure 5A and B. I would emphasize that these are minor issues which are important for the rigor of the published analysis but not the interpretation of the data. I expect that it will be fast to make any remaining changes and to clarify the open questions.

---

## [Author Response]

[Editors’ note: the authors resubmitted a revised version of the paper for consideration. What follows is the authors’ response to the first round of review.]

Essential revisions:1) The authors have good evidence that egg-derived phospholipids largely explain the transcriptional response to oviposition and at least two associated stress responses (SA accumulation, cell death) in the *P. brassicae*-*A. thaliana* interaction, but the evidence for the specific importance of egg-derived PCs is weaker. Specifically:1.1) PCs clearly do not account for the entire egg lipid-associated response, which the authors acknowledge and discuss.

We agree with the reviewers that the plant responses observed after PC treatment are less strong than the ones observed after EE treatment, which could be either due to additive effects from different PCs and other phospholipids in the EE or due to additional factors in the EE (lipid carriers, antioxidants, adjuvants). To clarify the role of PCs in the egg lipid-associated response we conducted two additional sets of experiments.

a) We compared the accumulation of *PR1* transcript levels in response to treatment with EE, a synthetic egg lipid mix (SELM), and a pure PC-Mix. The SELM was composed of the main phospholipids found in *P. brassicae* EE and at similar concentration. Moreover, we included a second SELM without the PC-Mix. Interestingly, treatment with EE, SELM, and PC-Mix induced the accumulation of *PR1* transcript levels but the SELM without PCs failed to induce a similar response in *Arabidopsis*. In our opinion this new experiment supports our conclusion that PCs are absolutely needed for the lipid-associated response that we observed in our experiments, with little or no contribution from other phospholipids, at least when the phospholipids are applied in concentrations comparable to the ones found in the EE. The new data are included in the manuscript in Figure 2G.

b) To test if other phospholipids (besides PCs) can induce a comparable response we treated plants with one synthetic phospholipid from each phospholipid class. We therefore chose phospholipids that were identified by MS analysis in *P. brassicae* EE. We applied the phospholipids at 1 µg/µl and measured *PR1* expression by qPCR. The data are included in the manuscript as a new supplement (Figure 3—figure supplement 1). PC(18:1/18:1) robustly activated *PR1* gene expression but in contrast LPC(18:1), PE(18:1/18:1), LPE(18:1), DAG(16:0/18:1), and TAG(18:1/18:1/18:1) were inactive. Interestingly, PA(18:1/18:1) activated *PR1* gene expression as well but to a lower extent than PC(18:1/18:1), which is discussed in the corresponding sections of the manuscript (subsection “PCs are active molecules in eggs” and Figure 5 legend). Notably, a PA concentration of 1 µg/µl is more than 50x higher than the PA concentration found in *P. brassicae* EE. When the PA-Mix was applied in a physiologically relevant concentration together with other EE phospholipids, but excluding PCs, the mixture was inactive (Figure 2G; SELM-PC).

1.2) Tests with purified PCs and PEs use chicken eggs as a source and the composition is not compared to the composition of PCs and PEs found in P. brassicae eggs.

We compared the total number of carbon atoms and the sum of the double bonds in the fatty acid acyl chains of the insect egg PCs and PEs with the fatty acid distribution of the chicken egg PC and PE mixes (Author response image 1). The majority of PCs and PEs in the insect EE contained C16 and C18 fatty acyl chains which were also the most abundant fatty acids in the PC and PE mixes purified from chicken eggs. We therefore concluded that the use of the PC and PE mixes was reasonable to test if treatment with these phospholipid classes can activate defense gene expression.

**Author response image 1. sa2fig1:** Comparison of chemical composition of PC and PE mixes purified from chicken egg yolk and PC and PE in EE. (A, B) Fatty acid distribution in the commercially available PC and PE mixes purified from chicken eggs used in this study (Redrawn from ‘Fatty Acid Distribution’ at: PC, https://avantilipids.com/product/840051; PE, https://avantilipids.com/product/840021). (C, D) Absolute levels of phosphatidylcholines (PCs) and phosphatidylethanolamines (PE) in *P. brassicae* and *S. littoralis* EE measured by MS shotgun lipidomic analysis. Different PC and PE species are reported according to their molecular composition with the total number of carbon atoms and the sum of double bonds in the fatty acyl chains. PC and PE levels represent the means ± SE of three different EE preparations.

1.3) PLA2 also cleaves other phospholipids, as does PLD (albeit with weaker activity), and thus the phospholipase assay demonstrates the importance of phospholipids but not PCs specifically; Figure 2F is misleading and the authors should show the full lipid spectrum.

We are aware that phospholipases can cleave other phospholipids than PCs as well. We changed Figure 2F according to the reviewer suggestion and adjusted the description and discussion in the text (subsection “Identification of active PCs”).

2) The authors need to test whether leaf-derived PCs (and perhaps other phospholipids, see 1) could also explain the response they observe, in which case the claim that these are "EAMPs" cannot be supported and this rather starts to look like a less specific damage-associated response.

Most of the phospholipids tested in Figure 3, Figure 3—figure supplement 1, and Figure 5 are synthetic compounds. We therefore argue that the chemical structure of those compounds will not be different based on their origin. A potential involvement of leaf-derived PCs as damage associated molecular patterns is discussed in the answer to the main comment 3 below.

3) The authors show that there are sufficient concentrations of PCs in eggs to elicit a response, but do not show that oviposited eggs actually release such concentrations to the leaf. This could in part be addressed by comparing the response of WT plants and LecRK-I.8 mutants to oviposition as well as to egg extract and an appropriate PC or phospholipid mix (again, see point 1).

We agree with the reviewers that a potential release of PCs from the insect eggs is not shown in the manuscript. To tackle this technically challenging question, we tested the release of PCs from eggs oviposited on filter paper. Butterflies can do that when exposed to leaves covered with paper. Strikingly, we could detect the presence of PCs on pieces of paper that contained *P. brassicae* and *S. littoralis* eggs for 1 to 3 days, and the profile was highly similar to the profile identified in EE (new Figure 4). The total amount of PCs released by *P. brassicae* eggs during one day was 0.23 ± 0.05 mg / batch, which falls in the range of active concentrations that induce *PR1* expression. These data thus strongly suggest that PCs can diffuse out of the eggs and are released in sufficient amounts to trigger responses in the plant.

In addition, we also tried to address this question with the suggested experiment. We thus compared the accumulation of SA and transcript levels of *PR1*, *SAG13*, and *TI* in Col-0 and *lecrk-I.8* in response to oviposition, EE, a PC-Mix and two synthetic PCs (PC(16:1/16:1); PC(18:3/18:3)). In addition, to clarify if the observed response could be triggered by leaf-derived PCs due to damage we included a localized wounding treatment. Leaves were wounded with a one-side corrugated forceps that wounded the leaves mostly from the abaxial site, which is the site of egg, EE, and PC application. The damaged area was similar to the area of EE or PC application (Author response image 2). Results indicate that oviposition, EE, and PC-treatment similarly induce a strong SA accumulation and upregulation of marker gene expression in Col-0 and that this response is significantly reduced in *lecrk-I.8*. In contrast, localized wounding does not trigger SA accumulation and marker gene expression. The new data are included in the manuscript in Figure 5A and B. These new experiments therefore support our conclusion that egg-derived PCs induce an immune response in *Arabidopsis* after oviposition and that this response involves the receptor kinase LecRK-I.8. Although we cannot formally exclude that wounding releases PCs from plant cells, our data suggest that this is not sufficient to trigger the responses observed after oviposition or after treatment with exogenous PCs. The results and discussion on potential leaf-derived PCs are now found in subsection “PC’s can diffuse out of the eggs” and the Discussion section of the revised manuscript.

**Author response image 2. sa2fig2:** Forceps used (left) and site of leaf wound treatment. The size of the corrugated wounding-surface was adjusted to the size of a 2 µl drop of EE. Dotted circle shows the wounded area.

4) Reviewer 1 requests that the authors document their statistical approach in the Materials and methods section (software used, testing that assumptions of statistical tests were met) and that the authors be ready to submit source data for all figures upon publication.

Statistical tests used are described in the Materials and methods section. A source data file for all figures is provided as an accompanying file.

5) Reviewer 2 points out that the elicited response in Arabidopsis has not been shown to defend against P. brassicae, and thus the authors should not refer to this as a defense response but might rather characterize it as an immune response.

We modified the title and text accordingly.

6) Reviewer 3 notes that phosphatidic acid (PA) was not tested, while the authors' own reasoning (shown in Figure 3B) indicates that this is relevant.

We share the concern of reviewer 3 that it is crucial to test further potential PC-related structures shown in Figure 3B. We therefore repeated that experiment three times with PC(18:1/181) and included commercially available LPC(18:1) and PA(18:1/18:1). We replaced Figure 3B with a new revised version. Treatment with the full PC(18:1/18:1) induced the expression of *PR1*, *SAG13*, and *TI*. Treatment with choline, phosphocholine, LPC(18:1), and oleic acid (C18:1) did not induce expression of the reporter genes. Consistent with the data shown in the new supplement to Figure 3 (Figure 3—figure supplement 1, please see the answer to main comment 1.1) treatment with PA(18:1/18:1) caused GUS staining on the reporter lines. We changed the text accordingly and included a part in the discussion which covers the role of PA in plant defense signaling and a potential conversion of exogenous PC to PA (subsection “PCs are active molecules in eggs” and Figure 5 legend).

We have included the individual peer reviews here for your information. However, please note that the comments above reflect the consensus of the reviewers and their common recommendation.Reviewer #1:[…]The authors, however, do not describe their statistical analyses in their methods section, and this needs to be rectified. See comments below. I would also encourage them to submit all source data.

Please see answer to the main comment 4 above.

Reviewer #2:[…] Although we find the study highly exciting and mostly elegantly performed, we have a few major concerns, namely:a) whether the PCs were the only active compounds in egg extract lipid fraction;

In the revised version of the manuscript we tested several other phospholipids found in EE, separately and in a synthetic egg lipid mix (Figure 3A and B; Figure 3—figure supplement 1; Figure 5; please see the answers to the main comments 1 and 6). In all experiments, the immune-response activation was consistently observed after PC treatment. We therefore conclude that phosphatidylcholines are the main active compound in the egg extract lipid fraction but we cannot formally exclude that other factors in the lipid fraction contribute to the induction of plant immune signaling, for example PAs (subsection “PCs are active molecules in eggs” and Figure 5 legend).

b) that the PCs shown to be bioactive were actually not purified from the butterfly eggs (as claimed in the title and in the Discussion section);

Unfortunately, due to their high chemical similarity, a further fractionation and purification of the insect egg PCs is not possible with the methods and analytical machines currently available. However, as most of the phospholipids tested in Figure 3, Figure 3—figure supplement 1, and Figure 5 are synthetic compounds we argue that the chemical structure of those compounds will not differ based on their origin. In addition, PC (16:1/16:1) and PC (18:3/18:3) were formally identified by MS in the EE, and both compounds activate EE responses when applied on the plant.

c) that plant-derived PCs have not been tested in order to rule out whether they are likely to induce a similar response (we would need to call them then DAMPs (damage-associated molecular patterns) instead of EAMPs);

As mentioned above, the structure of plant-derived PCs should not be different from the synthetic ones used in this study. A potential role of leaf-derived PCs as damage associated molecular patterns is addressed in the revised version of Figure 4 (now Figure 5), please see the answer to the main comment 3 above. Information is detailed in subsection “PC’s can diffuse out of the eggs” and the Discussion section of the revised manuscript.

d) whether the bioactivity of PC (16:1/16:1) is dependent on the whole molecule since lisophosphatidylcholine (released with a fatty acid by PLA2-mediated PC cleavage) and phosphatic acid (released with a choline by PLD-mediated cleavage) have been previously shown to be bioactive in plants and;

Figure 3B has been revised with PC (18:1/18:1) and we included commercially available LPC(18:1) and PA(18:1/18:1). Please see the answer to the main comment 6 above.

e) that the study might not be novel enough (as claimed in the abstract) as similar EAMPs were found in planthopper eggs.

PCs from planthopper were purified from adult females and their specific origin was not shown. Furthermore, no molecular and signaling information was provided.

Not all lipids present in the PR1-inducing subfractions 4.10-4.17 were tested, like SMs (sphingomyelins) and PEs (phosphatidylcholines). These two appear to be present in most of the PR1-active fractions but were not further evaluated for their activity as pure compounds (except the PE mix from chicken eggs). The authors motivate the focus on PCs based on the PLD + PLA2-treated egg extract that lose PR1 induction (Figure 2E). PLD and PLA2 could be active also on PEs, thus PEs should still be tested as candidates. Furthermore, Figure 2F should also show the effect of PLD+ PLA2 across the whole lipid NMR spectra not only on PCs. The ultimate proof that insect-derived PCs are active, should be done by testing PCs purified form the HPLC bioactive subfractions 4.10-4.17. Assuming the activity of insect egg derived PCs, the questions on their origin is still unresolved, given their ubiquity in plant cells.

We agree with the concern of reviewer 2 that SMs were not tested and that the activity of PEs was just tested with a PE mix purified from chicken eggs. We therefore tested a SM-Mix for its ability to induce the expression of *PR1*, *SAG13*, and *TI* and included the new data in Figure 3A. Treatment with the SM-Mix did not cause any GUS staining on the used reporter lines. Moreover, synthetic PE(18:1/18) and LPE(18:1) did not induce *PR1* (Figure 3—figure supplement 1; please see the answer to main comment 1.1b).

Figure 2F was changed according to the reviewer's suggestion (please see the response to main comment 1.3).

As mentioned above a further fractionation and purification of the insect eggs PCs is not possible with the methods and analytical machines available to us and we therefore chose synthetic PCs with similar chemical structures than the ones identified in the EE.

We ruled out the possibility that active PCs responsible for egg-induced responses originate from plant cells (as DAMPs) by testing localized wounding. See response to main comment 3. Information detailed in subsection “PC’s can diffuse out of the eggs” and the Discussion section of the revised manuscript.

Lastly, the authors claim that their results point to the conservation of phospholipid EAMPs between distantly related insect species, including moths, planthoppers or true flies. However, most eukaryotes incl. plant membranes and some bacteria contain PCs and, in our view, it cannot be stated yet that this immunity response is specific to insect eggs. And that brings us to our last major concern that *A. thaliana* is unlikely to have evolved such a response to defend insect eggs: neither is it a host plant for one of the tested lepidopteran species nor has it been shown to actually defend eggs, by e.g. impeding egg survival.

The reviewer correctly points that PCs cannot be specific to insect eggs. We reformulated the sentence (Discussion section).

Previous studies have shown that plants from the Brassicaceae (including *Arabidopsis*) respond to *P. brassicae* oviposition with localized necrosis, SA accumulation and defense gene expression (Bruessow and Reymond, 2007; Bruessow et al., 2010; Fatouros et al., 2014; Bonnet et al., 2017; Griese et al., 2017; Lortzing et al., 2020). In *Brassica nigra*, variation in the intensity of localized cell death was negatively correlated with egg survival (Fatouros et al., 2014; Griese et al., 2017). In *Arabidopsis*, we have observed variation in the strength of egg-induced necrosis between accessions, suggesting some ecological relevance for this response (Reymond, 2013). Preliminary observations with accessions showing strong necrosis indicate that *P. brassicae* hatching rate is reduced. We thus believe that *Arabidopsis* may have evolved a defense mechanism against eggs from Pierids or other lepidopteran herbivores, but this has to be further studied in the right ecological context. We added more information in the Introduction.

Reviewer #3:[…]However, as a major concern from the side of this reviewer, this conclusion seems not justified since the authors provide no direct evidence that the response to oviposition indeed proceeds via (substantial) release of PC from the intact insect eggs on the leaf surface. A first line of evidence supporting such a process could consist in experiments demonstrating that the response to oviposition is equally dependent on LecRK-I.8 as the response to PC and EE. Such evidence is missing in the current work and seems to be missing also in the previous publications on LecRK-I.8 (at least I could not detect such data on first glance in the publications available to me).An experimentally confirmed activation mechanism with Arabidopsis reacting to oviposition via (direct or indirect) detection of such a common molecule as PC would certainly add a new and intriguing aspect as to how plants can detect adverse changes in their environment.

We agree with reviewer 3 that it is crucial to compare of Col-0 and *lecrk-I.8* in response to oviposition, EE, a PC-Mix and pure PCs. We included such experiments in the revised version of the manuscript (new Figure 5), please see the answer to main comment 3 above.

[Editors’ note: what follows is the authors’ response to the second round of review.]

The reviewers agree that the revisions have fully addressed the concerns expressed in the prior round of review and unanimously recommend acceptance; however, three revisions are required before the article is formally:1) Authors should briefly explain the two different nomenclatures used to refer to phospholipids in order to avoid reader confusion, as I expect that readers with a background in molecular biology or plant-insect interactions may not immediately realize the parity (e.g. 36:2 versus 18:1/18:1). This can be quickly done in the explanations of Figure 2 and Figure 3.

We have included a sentence describing the PC nomenclature used in this paper (lines 148-152).

2) Variation in egg-induced necrosis in *A. thaliana* has been reported suggesting "some ecological relevance in the model species" (Introduction).It may still be too early to draw such a conclusion in absence of reports on A. thaliana expressing a necrosis sufficient to reduce egg survival, considering also that A. thaliana has not yet been reported as a natural host for P. brassicae. Please revise wording.

We have changed the sentence accordingly (Introduction).

3) The authors should note whether and how assumptions of their chosen statistical tests were evaluated (homogeneity of variance, normal distribution) and whether data were transformed prior to analysis if these assumptions were not met.

We have carefully reevaluated our data and applied the appropriate statistical tests. We added this information in the Material and Methods section and Figure legends.

[Editors’ note: what follows is the authors’ response to the third round of review.]

We are ready to accept your manuscript conditional on a few additional changes which, however, should be fast to make.The last round of review requested that the authors state how they determined the suitability of statistical tests (e.g. homogeneity of variance, normal distribution) and whether any data were transformed. The authors have added corresponding statements to the text but these do not always seem to fit with the data as shown in the figures. In general, it looks as though it would be necessary to either log-transform or use a nonparametric test for most of the quantitative transcript accumulation data due to the very large differences in variance as a result of the control levels being near the limit of detection (e.g. Figure 1B, Figure 2 E and G). It is also strange that the same problems with homogeneity of variance were not encountered in both Figure 5 A and B. I would emphasize that these are minor issues which are important for the rigor of the published analysis but not the interpretation of the data. I expect that it will be fast to make any remaining changes and to clarify the open questions.

Following your suggestion, we have done the statistical analyses on log-transformed data for gene expression analyses. We also used the Welch's t-test for pair-wise comparisons to account for large differences in variance between controls (close to limit of detection) and treatments. This information is included in the Materials and methods section.